# Prevalence and associated risk factors of tinnitus among Palestinian adolescents aged 15–18: A cross-sectional study

Saad Al-Lahhaam[1]*, Raghad Dweikat[2], Tala Nazzal[2], Aman Maraqa[2], Joud Khalil[2], Tala Albadawi[2], Raghad Doufish[2], Wa'd Amer[2], Mustafa Ghanim[1], Mohammad Abuawad[1], Amer Ghrouz[2], Samar Alkhaldi[2], Laith El-lahham[3], Majdi Dwikat[2], Maha Rabayaa[1], Malik Alqub[1]*

1 Department of Biomedical Sciences and Basic Clinical Skills, Faculty of Medicine and Allied Medical Sciences, An-Najah National University, Nablus, Palestine, 2 Department of Allied and Applied Medical Sciences, Faculty of Medicine and Allied Medical Sciences, An-Najah National University, Nablus, Palestine, 3 Faculty of Medicine, Al-Quds University, Jerusalem, Palestine

* saedallahham@gmail.com (SAL); m.alqub@najah.edu (MA)

## Abstract

### Background

Tinnitus is a prevalent condition worldwide, particularly among adolescents, that has a substantial impact on quality of life, yet it remains an understudied issue.

### Objectives

This study aims to determine the prevalence of tinnitus and its associated risk factors among Palestinian adolescents aged 15–18.

### Methods

A cross-sectional study was conducted from January to March 2025. A convenience sample of participants was recruited. The study utilized the European School for the Interdisciplinary Tinnitus Research Screening Questionnaire.

### Results

A total of 1,131 participants were enrolled in the study, with 64.5% being females. The prevalence of tinnitus among the study sample was 532, representing 47% of the population. Females had a higher prevalence of tinnitus, with 370 affected (50.7%) compared to males (40.4%). Significant associations were found between tinnitus and several factors: age, positive family history of tinnitus (threefold increased risk), sensitivity to external sounds (2.7 times higher likelihood), slight hearing difficulty in noisy environments (1.7 times higher risk), pain symptoms (double the risk), and difficulty falling asleep (1.8 times higher risk). Notably, the majority of affected participants (71.5%) had never sought professional care for their tinnitus.

**Data availability statement:** All relevant data are within the manuscript and its Supporting Information files.

**Funding:** The author(s) received no specific funding for this work.

**Competing interests:** The authors have declared that no competing interests exist.

## Conclusion

Although Tinnitus is common among Palestinian adolescents aged 15–18 years, the majority of affected participants did not seek professional care for tinnitus. These findings highlight the importance of conducting further research to shed insight into this prevalent and neglected health priority.

## Introduction

Tinnitus is described as the perception of sound in the absence of external auditory stimuli. Commonly, individuals with tinnitus report hearing sounds such as hissing, sizzling, or ringing. More complex auditory experiences, such as voices or music, have also been documented in some patients [1]. Tinnitus that coincides with the heartbeat may suggest a vascular origin [2,3]. When it does not, palatal muscles or myoclonus of the middle ear is more likely to be the cause, while a lack of such synchronization often points to other causes, such as palatal muscle spasms or myoclonus of the middle ear [2]. Tinnitus typically develops gradually, although it can sometimes have a rapid onset [4]. Many patients report that stress can exacerbate their symptoms, highlighting the potential influence of emotional and psychological factors in the experience of tinnitus [5].

Despite extensive research, the mechanisms behind tinnitus remain poorly understood. It is believed to arise from complex interactions between the auditory pathways, the central nervous system, and other systems. Several factors have been identified as potential risk factors for the development of tinnitus. These include otological (hearing-related), neurological, psychiatric, cardiovascular, traumatic, rheumatological, immune-mediated, endocrine, and metabolic factors. Lifestyle factors such as previous head injuries, alcohol consumption, arthritis, obesity, smoking, and hypertension have also been identified as contributing to the risk of tinnitus [6–9]. The challenges in tinnitus research stem from its multifactorial etiology, the diverse comorbidities associated with it, its wide clinical variability, and the subjective nature of its assessment [10,11].

Tinnitus is a widespread condition, affecting a significant proportion of the global population [12]. Significant variation in the prevalence of tinnitus was reported from numerous studies across Middle Eastern countries. Estimates of tinnitus prevalence in the general population have been reported at 30.6% in Palestine [7], 33% in Turkey [13], 5.2% in Egypt [14], and 4.6% in Iran [15]. The prevalence of tinnitus in children has been reported to range from 7.5% to 60% [12,16]. Tinnitus is believed to be more common in children with hearing impairment compared with children with normal hearing [17].

In Palestine, the prevalence of tinnitus has been estimated at 30.6% among adults [7] and 31% among university students [18]. Yet data on Palestinian adolescents remain scarce. This represents a critical gap, as tinnitus is increasingly recognized as a condition with significant psychosocial dimensions beyond its auditory characteristics [19]. In adolescents, tinnitus may be closely intertwined with psychological stress,

anxiety, sleep disturbance, and daily functioning [20,21], particularly during a developmental stage characterized by academic pressure and heightened emotional sensitivity [22]. These impacts can substantially affect quality of life, emotional equilibrium, and social engagement [23,24]. Conceptualizing tinnitus within a biopsychosocial framework is therefore essential to understanding its mechanisms and risk factors in young populations, as well as for addressing its potential effects on their quality of life [25]. Therefore, the current study aims to examine the prevalence and risk factors of tinnitus among Palestinian children aged 15–18 years.

## Methods

### Study design and population

A cross-sectional questionnaire-based study was conducted from January to March 2025 to assess the prevalence of tinnitus and its associated risk factors.

The sample size was calculated using Cochran's formula for prevalence studies:

$$n = (Z^2 \times p \times (1 - p)) / e^2$$

Assuming a 95% confidence level (Z = 1.96), an expected prevalence of 50% (p = 0.5; no prior regional data), and a 5% margin of error (e = 0.05), the initial sample size was determined to be 385 participants. Given the large target population (N ≈ 340,000), the finite population correction had a negligible impact, and the sample size remained unchanged. To account for potential non-response, the target was increased by 15%, resulting in a final sample size of 443 participants.

### Study setting, sampling method, and data collection procedure

The study involved a population survey of Palestinians aged from 15 to 18 years. The population included individuals from all four governorates of the West Bank in Palestine. The participants in this study were recruited from different settings across the West Bank in Palestine to ensure a representative sample. Participants were recruited by selecting individuals from local community centers and social media platforms. The questionnaire was shared with people and friends, including researchers' accounts, to reach a larger number of community participants, enabling a deeper understanding of the study's findings.

Given the contextual limitations in Palestine, such as restricted institutional access, lack of centralized databases for adolescents, and logistical barriers due to political instability, probability-based sampling strategies were not feasible. Therefore, a large-scale convenience sampling approach was employed, which is a commonly used method in public health research in low-resource or conflict-affected settings. This approach allowed us to efficiently collect data from a broad and diverse sample and generate early epidemiological insights on tinnitus, a condition for which there is limited evidence in this population.

### Study tools

The study's questionnaire was based on the European School for Interdisciplinary Tinnitus Research-Screening Questionnaire (ESIT-SQ), a self-report tinnitus-relevant history questionnaire (supplementary file) [23]. Public health and speech disorders professionals examined and updated the questionnaire before distribution. The questionnaire comprised questions addressing the sociodemographic data, clinical characteristics of the participants, the characteristics of tinnitus, inciting triggers, treatment approaches, and responses to therapy among the tinnitus-affected participants. The questionnaire was an Arabic translation version of ESIT-SQ, which was used as a study tool in a recent Palestinian studies, the questionnaire underwent a forward-backward translation process to ensure language accuracy in prior research [7,18]. The study questionnaire was distributed digitally through email, institutional platforms, etc. Participation was entirely voluntary. A pilot study was executed before the comprehensive data collection to evaluate the clarity, relevance, and feasibility of

the questionnaire. The pilot research data were excluded from the final analysis. The study was conducted using a digital questionnaire created via Google Forms. At the beginning of the form, an electronic informed consent statement was presented. Only participants who clicked "I agree" were able to proceed with the rest of the questionnaire. This ensured that informed consent was obtained digitally and voluntarily before any data were collected. To minimize respondent fatigue, the survey platform allowed participants to pause and resume, reducing the likelihood of inconsistent responses toward the end. A total of 1131 responses were received from participants who agreed and completed the questionnaire.

### Data analysis

The statistical analyses were carried out using IBM Corp.'s Statistical Package for the Social Sciences version 25 (SPSS 25) in Armonk, New York, USA. Data was analyzed descriptively, with categorical variables expressed as frequency and percentage. To examine associations between tinnitus and various socio-demographic and clinical variables, Chi-square tests or Fisher's exact tests (as appropriate, based on cell counts) were performed for bivariate comparisons. A p-value of < 0.05 was considered statistically significant. Variables that showed significant associations in the bivariate analysis were subsequently entered into a binary multivariate logistic regression model to identify independent predictors of tinnitus. In addition, the association between the subtypes of tinnitus (objective vs. subjective tinnitus, continuous vs. intermittent tinnitus) and the characteristics of participants with tinnitus was evaluated using the chi-square and, followed by the binary logistic regression for significant variables to determine the predictive independent variables. The enter method was used for variable entry. Results were reported as adjusted odds ratios (ORs) with 95% confidence intervals (CIs).

### Ethical consideration

The Institutional Review Board (IRB) of An-Najah University provided its approval for this study, which was conducted in accordance with ethical standards and was performed in compliance with the Helsinki Declaration for research in humans (Reference: BioMed. Dec.2024/23). Before inclusion, all participants provided written informed consent to take part in the study. All participants provided informed consent before proceeding to the online questionnaire questions, and those who did not sign the consent form were not allowed to participate. The permission form explained the study's goal and guaranteed voluntary and anonymous participation, with no consequences for nonparticipation.

## Results

### The study population

The survey included 1131 participants. 35.5% were male, and 64.5% were female. The participants' ages ranged from 15 to 18 years, with 30.2% being 15 years old, 32.2% being 16 years old, 28.2% being 17 years old, and 9.4% being 18 years old. The majority of the participants had a normal BMI (66%), with 16.5% underweight, 13.5% overweight, and 3.9% obese. Regarding city of residence, 37.3% were from Hebron, 25.2% from Nablus, 30.6% from Ramallah, and 6.9% from Bethlehem. 62% of participants reported high phone usage. The majority of participants (85.9%) were right-handed. Among the participants, 8.5% reported having a mother with tinnitus, 5% a father, 3.9% brothers, and 4% sisters with tinnitus. 46.7% of the participants didn't know if they had first-degree relatives with tinnitus, while 40.4% reported having no relatives with tinnitus. Among those who reported having relatives with tinnitus, the mean number of affected relatives was 1.92 ± 1.292. **Table 1** displays the participants' characteristics.

### Clinical characteristics of the participant

**Table 2** presents the clinical features of the study participants. Regarding dizziness, 42.7% reported experiencing it more than once a year, 26.1% less than once a year, and 31.2% reported no dizziness. Among ear-related conditions, acoustic trauma due to sudden loud noise affected 3.6%, and middle ear infection due to external pressure was present in 8.4%. Notably,

**Table 1. Background characteristics of the study participants, descriptive statistics of the sample characteristics (*n* = 1131).**

| Variable | Frequency | Percentage (%) |
|---|---|---|
| Age | | |
| **15** | 342 | 30.2% |
| **16** | 364 | 32.2% |
| **17** | 319 | 28.2% |
| **18** | 106 | 9.4% |
| Gender | | |
| **Male** | 401 | 35.5% |
| **Female** | 730 | 64.5% |
| BMI | | |
| **Underweight >18.5** | 187 | 16.5% |
| **Normal weight 18.524.9** | 746 | 66% |
| **Overweight 2529.9** | 153 | 13.5% |
| **Obesity ≥ 30** | 44 | 3.9% |
| City | | |
| **Hebron** | 422 | 37.3% |
| **Nablus** | 285 | 25.2% |
| **Ramallah** | 346 | 30.6% |
| **Bethlehem** | 78 | 6.9% |
| High Phone Usage | | |
| **Yes** | 701 | 62% |
| **No** | 56 | 5% |
| **Not much** | 374 | 33.1% |
| Dominant Writing Hand | | |
| **Right** | 971 | 85.9% |
| **Left** | 64 | 5.7% |
| **Both** | 96 | 8.5% |
| First-Degree Relatives with Tinnitus | | |
| **Mother** | 96 | 8.5% |
| **Father** | 57 | 5% |
| **Brothers** | 44 | 3.9% |
| **Sisters** | 45 | 4% |
| **I don't know** | 528 | 46.7% |
| **None** | 457 | 40.4% |
| Number of Affected Relatives | | |
| **≤ 3** | 121 | 10.7% |
| **4–6** | 21 | 1.9% |
| **≥ 7** | 1 | 0.1% |
| Number of Affected Relatives (Mean ± SD) | **1.92 ±1.292** | |

age-related hearing loss, acoustic neuroma, otosclerosis, and sudden hearing loss were rare, each affecting less than 1% of participants. A significant majority, 82.6%, reported no ear-related conditions. In terms of prior medical procedures, 20.2% had undergone dental procedures, and 2.3% had ear surgery, while a substantial 74.8% reported no prior medical procedures. Sensitivity to external sounds in the past week was observed in 34.7% of participants. Regarding hearing difficulties, 42.4%

**Table 2. Clinical characteristics of the study participants (*n* = 1131).**

| Variable | Frequency | Percentage (%) |
|---|---|---|
| Dizziness | | |
| **Yes, less than once a year** | 295 | 26.1% |
| **Yes, more than once a year** | 483 | 42.7% |
| **No** | 353 | 31.2% |
| Ear-related conditions: | | |
| Acoustic trauma due to sudden loud noise | | |
| **Yes** | 41 | 3.6% |
| **No** | 1090 | 96.4% |
| Middle ear infection due to external pressure | | |
| **Yes** | 95 | 8.4% |
| **No** | 1036 | 91.6% |
| Age-related hearing loss | | |
| **Yes** | 1 | 0.1% |
| **No** | 1130 | 99.9% |
| Sudden hearing loss | | |
| **Yes** | 7 | 0.6% |
| **No** | 1124 | 99.4% |
| Acoustic neuroma | | |
| **Yes** | 1 | 0.1% |
| **No** | 1130 | 99.9% |
| Chronic ear infection | | |
| **Yes** | 27 | 2.4% |
| **No** | 1104 | 97.6% |
| Otosclerosis | | |
| **Yes** | 1 | 0.1% |
| **No** | 1130 | 99.9% |
| Middle ear infection or Eustachian tube dysfunction | | |
| **Yes** | 25 | 2.2% |
| **No** | 1106 | 97.8% |
| Eardrum perforation | | |
| **Yes** | 20 | 1.8% |
| **No** | 1111 | 98.2% |
| Hearing loss due to other reasons | | |
| **Yes** | 17 | 1.5% |
| **No** | 1114 | 98.5% |
| None conditions | | |
| **Yes** | 934 | 82.6% |
| **No** | 197 | 17.4% |
| Prior medical procedures: | | |
| Ear surgery | | |
| **Yes** | 26 | 2.3% |
| **No** | 1105 | 97.7% |
| Dental procedure (filling removal, dental implants, prolonged dental surgery) | | |
| **Yes** | 228 | 20.2% |
| **No** | 903 | 79.8% |
| Neurosurgery | | |

*(Continued)*

| Variable | Frequency | Percentage (%) |
|---|---|---|
| **Yes** | 13 | 1.1% |
| **No** | 1118 | 98.9% |
| Spinal tap (lumbar puncture) | | |
| **Yes** | 10 | 0.9% |
| **No** | 1121 | 99.1% |
| Chemotherapy | | |
| **Yes** | 5 | 0.4% |
| **No** | 1126 | 99.6% |
| Radiation therapy for the head and neck | | |
| **Yes** | 11 | 1% |
| **No** | 1120 | 99% |
| Electroconvulsive therapy | | |
| **Yes** | 5 | 0.4% |
| **No** | 1126 | 99.6% |
| Other; Nasal polypectomy | | |
| **Yes** | 4 | 0.4% |
| **No** | 1127 | 99.6% |
| Other; Tonsillectomy | | |
| **Yes** | 17 | 1.5% |
| **No** | 1114 | 98.5% |
| Other; Pulmonary Laceration | | |
| **Yes** | 1 | 0.1% |
| **No** | 1130 | 99.9% |
| Other; Hand Fracture | | |
| **Yes** | 1 | 0.1% |
| **No** | 1130 | 99.9% |
| Other: Sleeve Gastrectomy | | |
| **Yes** | 1 | 0.1% |
| **No** | 1130 | 99.9% |
| None procedures | | |
| **Yes** | 846 | 74.8% |
| **No** | 285 | 25.2% |
| Sensitivity to External Sounds during the past week | | |
| **Yes** | 392 | 34.7% |
| **No** | 739 | 65.3% |
| Difficulty Hearing in Noisy Environments | | |
| **No** | 619 | 54.7% |
| **Slight difficulty** | 479 | 42.4% |
| **Yes, I cannot hear at all** | 33 | 2.9% |
| Hearing Assistive Device Usage: | | |
| Hearing aid | | |
| **Yes** | 8 | 0.7% |
| **No** | 1123 | 99.3% |
| Cochlear implant | | |
| **Yes** | 1 | 0.1% |

*(Continued)*

Table 2. (Continued)

| Variable | Frequency | Percentage (%) |
|---|---|---|
| **No** | 1130 | 99.9% |
| Sound generator | | |
| **Yes** | 2 | 0.2% |
| **No** | 1129 | 99.8% |
| Combined device (hearing aid and sound generator in one device) | | |
| **Yes** | 3 | 0.3% |
| **No** | 1128 | 99.7% |
| None devices | | |
| **Yes** | 14 | 1.2% |
| **No** | 1117 | 98.8% |
| Experience of Symptoms: | | |
| Headache | | |
| **Yes** | 474 | 41.9% |
| **No** | 657 | 58.1% |
| Neck pain | | |
| **Yes** | 208 | 18.4% |
| **No** | 923 | 81.6% |
| Ear pain | | |
| **Yes** | 112 | 9.9% |
| **No** | 1019 | 90.1% |
| Jaw joint pain | | |
| **Yes** | 101 | 8.9% |
| **No** | 1030 | 91.1% |
| Facial pain | | |
| **Yes** | 41 | 3.6% |
| **No** | 1090 | 96.4% |
| Other; Shoulder pain | 5 | 0.4% |
| **Yes** | 1126 | 99.6% |
| **No** | | |
| None Pain | | |
| **Yes** | 573 | 50.7% |
| **No** | 558 | 49.3% |
| Oral conditions: | | |
| Jaw joint pain | | |
| **Yes** | 106 | 9.4% |
| **No** | 1025 | 90.6% |
| Dental problems | | |
| **Yes** | 357 | 31.6% |
| **No** | 774 | 68.4% |
| Neurological conditions: | | |
| Meningitis | | |
| **Yes** | 11 | 1% |
| **No** | 1120 | 99% |
| Multiple sclerosis | | |
| **Yes** | 8 | 0.7% |

*(Continued)*

**Table 2.** (Continued)

| Variable | Frequency | Percentage (%) |
|---|---|---|
| **No** | 1123 | 99.3% |
| Epilepsy | | |
| **Yes** | 24 | 2.1% |
| **No** | 1107 | 97.9% |
| Stroke | | |
| **Yes** | 3 | 0.3% |
| **No** | 1128 | 99.7% |
| Other cerebrovascular diseases | | |
| **Yes** | 9 | 0.8% |
| **No** | 1122 | 99.2% |
| Psychological conditions: | | |
| Anxiety, excessive stress | | |
| **Yes** | 426 | 37.7% |
| **No** | 705 | 62.3% |
| Depression | | |
| **Yes** | 141 | 12.5% |
| **No** | 990 | 87.5% |
| Emotional trauma | | |
| **Yes** | 82 | 7.3% |
| **No** | 1049 | 92.7% |
| Sleep disorders: | | |
| Difficulty falling asleep | | |
| **Yes** | 372 | 32.9% |
| **No** | 759 | 67.1% |
| Difficulty staying asleep | | |
| **Yes** | 180 | 15.9% |
| **No** | 951 | 84.1% |
| Cardiovascular conditions: | | |
| Low blood pressure | | |
| **Yes** | 54 | 4.8% |
| **No** | 1077 | 95.2% |
| High blood pressure | | |
| **Yes** | 35 | 3.1% |
| **No** | 1096 | 96.9% |
| Myocardial infarction (heart attack) | | |
| **Yes** | 2 | 0.2% |
| **No** | 1129 | 99.8% |
| Endocrine and metabolic conditions: | | |
| Thyroid dysfunction | | |
| **Yes** | 24 | 2.1% |
| **No** | 1107 | 97.9% |
| Diabetes | | |
| **Yes** | 10 | 0.9% |
| **No** | 1121 | 99.1% |
| High cholesterol | | |

*(Continued)*

**Table 2.** (Continued)

| Variable | Frequency | Percentage (%) |
|---|---|---|
| **Yes** | 11 | 1% |
| **No** | 1120 | 99% |
| Rheumatic and autoimmune disorders: | | |
| Rheumatoid arthritis | | |
| **Yes** | 22 | 1.9% |
| **No** | 1109 | 98.1% |
| Lupus (facial rash) | | |
| **Yes** | 9 | 0.8% |
| **No** | 1122 | 99.2% |
| Ear, nose, and throat conditions: | | |
| Chronic sinusitis | 117 | 10.3% |
| **Yes** | 1014 | 89.7% |
| **No** | | |
| Deviated nasal septum | 46 | 4.1% |
| **Yes** | 1085 | 95.9% |
| **No** | | |
| Other conditions: | | |
| Anemia | | |
| **Yes** | 80 | 7.1% |
| **No** | 1051 | 92.9% |
| Balance disorder | | |
| **Yes** | 79 | 7% |
| **No** | 1052 | 93% |
| Gastroesophageal reflux disease (GERD) | | |
| **Yes** | 27 | 2.4% |
| **No** | 1104 | 97.6% |
| Adenoid | | |
| **Yes** | 2 | 0.2% |
| **No** | 1129 | 99.8% |
| Asthma | | |
| **Yes** | 3 | 0.3% |
| **No** | 1128 | 99.7% |
| Epistaxis | | |
| **Yes** | 2 | 0.2% |
| **No** | 1129 | 99.8% |
| Familial Mediterranean Fever | | |
| **Yes** | 1 | 0.1% |
| **No** | 1130 | 99.9% |
| HSV | | |
| **Yes** | 1 | 0.1% |
| **No** | 1130 | 99.9% |
| Helicobacter pylori | | |
| **Yes** | 1 | 0.1% |
| **No** | 1130 | 99.9% |

*(Continued)*

**Table 2.** (Continued)

| Variable | Frequency | Percentage (%) |
|---|---|---|
| Iron Deficiency | | |
| Yes | 1 | 0.1% |
| No | 1130 | 99.9% |
| Leukemia | | |
| Yes | 1 | 0.1% |
| No | 1130 | 99.9% |
| Migraine | | |
| Yes | 5 | 0.4% |
| No | 1126 | 99.6% |
| OCD | | |
| Yes | 1 | 0.1% |
| No | 1130 | 99.9% |
| Schizophrenia | | |
| Yes | 1 | 0.1% |
| No | 1130 | 99.9% |
| Parkinson's disease | | |
| Yes | 1 | 0.1% |
| No | 1130 | 99.9% |

reported slight difficulty hearing in noisy environments, and 2.9% reported complete hearing loss. The use of hearing assistive devices was uncommon, with only 0.7% using hearing aids and even smaller percentages using cochlear implants, sound generators, or combined devices. Headache was a prevalent symptom, affecting 41.9% of the participants. Psychological conditions were also common, with 37.7% experiencing anxiety or excessive stress. Sleep disturbances were present in 32.9% of participants who reported difficulty falling asleep. In contrast, several conditions demonstrated low prevalence within the participant group. For instance, within the cardiovascular conditions, myocardial infarction (heart attack) was reported by only 0.2% of participants. Among other conditions, Familial Mediterranean Fever, HSV (Herpes Simplex Virus), Helicobacter pylori, Iron Deficiency, Leukemia, OCD (obsessive-compulsive disorder), Schizophrenia, and Parkinson's disease each affected a mere 0.1%. Detailed descriptions of these clinical characteristics are presented in **Table 2**.

### Prevalence of tinnitus and its characteristics

Of the participants affected by tinnitus (n = 532), 12.6%, 17.9%, and 13.5% experienced it daily or almost daily, weekly, and monthly, respectively. The majority of participants (28.2%) experienced tinnitus every few months, with 27.8% experiencing it yearly. 32.7% of those affected reported constant tinnitus, while 67.3% reported intermittent tinnitus. Tinnitus sounded like a cricket in 34.8% of the affected participants, and 27.6% perceived it as tonal. Tinnitus was perceived inside the head by 25.6% of the participants. The majority of tinnitus cases (74.1%) had no rhythm. At least 6% of individuals reported tinnitus audible to the clinician. The results are presented in **Table 3**.

As presented in **Table 4**, for many, tinnitus onset was unclear, with 57.7% not knowing when it began. In a small number of cases, tinnitus occurred before (2.6%), after (4.3%), or around the same time (1.1%) as other related conditions. Prominent conditions associated with tinnitus onset included exposure to loud sounds (28.2%), flu, common cold, or other infections (21.6%), and anxiety (20.9%). Concerning drug intake among respondents, 16.7% reported using pain relievers, while smaller percentages reported using antibiotics (9.2%), aspirin (1.9%), or quinine (1.9%). In terms of management, **Table 5** shows that the majority of participants didn't seek professional help, with 71.5% reporting they had never seen a healthcare professional for their tinnitus.

**Table 3. Characteristics of tinnitus among the affected participants (*n* = 532).**

| Variable | Frequency | Percentage (%) |
|---|---|---|
| Experience of tinnitus in one or both ears lasting more than 5 minutes at a time over the past year | | |
| **Yes, most or all of the time** | 26 | 2.3% |
| **Yes, frequently** | 66 | 5.8% |
| **Yes, occasionally** | 214 | 18.9% |
| **No, not in the past year** | 49 | 4.3% |
| **Never** | 550 | 48.6% |
| **I don't know** | 226 | 20% |
| Frequency of tinnitus (n = 532) | | |
| **Daily or almost daily** | 67 | 12.6% |
| **About weekly** | 95 | 17.9% |
| **About monthly** | 72 | 13.5% |
| **Every few months** | 150 | 28.2% |
| **Yearly** | 148 | 27.8% |
| Description of tinnitus | | |
| **Constant** | 174 | 32.7% |
| **Intermittent** | 358 | 67.3% |
| Tinnitus sound like | | |
| **Tonal (continuous sound with varying frequencies)** | 147 | 27.6% |
| **Noise-like** | 142 | 26.7% |
| **Music-like** | 31 | 5.8% |
| **Sounds like a cricket** | 185 | 34.8% |
| **Buzzing** | 20 | 3.8% |
| **Other; I don't know** | 2 | 0.4% |
| **Other; None** | 5 | 0.9% |
| Tinnitus rhythm | | |
| **Yes, it follows my heartbeat (it may be checked by feeling the pulse at the same time as the tinnitus)** | 45 | 8.5% |
| **Yes, it follows my breathing** | 34 | 6.4% |
| **Yes, it follows head, neck, jaw, or facial muscle movements** | 53 | 10% |
| **Other; I don't know** | 6 | 1.1% |
| **Other; None** | 394 | 74.1% |
| Tinnitus perceived | | |
| **Right ear** | 67 | 12.6% |
| **Left ear** | 47 | 8.8% |
| **Both ears, worse in the right** | 60 | 11.3% |
| **Both ears, worse in the left ear** | 22 | 4.1% |
| **Both ears equally** | 136 | 25.6% |
| **Inside the head** | 73 | 13.7% |
| **Other; None** | 4 | 0.8% |
| **I don't know** | 123 | 23.1% |
| Audible by the clinician | | |
| **Yes** | 32 | 6% |
| **No** | 500 | 94% |

**Table 4. Onset of tinnitus-associated conditions and respondent drug intake (*n*=532).**

| Variable | Frequency | Percentage (%) |
|---|---|---|
| Onset of Tinnitus | | |
| **1–12 months** | 98 | 18.4% |
| **13–24 months** | 72 | 13.5% |
| **25–36 months** | 31 | 5.9% |
| **> 36 months** | 14 | 2.6% |
| **I don't know** | 307 | 57.7% |
| Timing of previously mentioned conditions or procedures relative to tinnitus | | |
| **Before tinnitus started** | 14 | 2.6% |
| **After tinnitus started** | 23 | 4.3% |
| **Around the same time, tinnitus started** | 6 | 1.1% |
| **I don't know** | 17 | 3.2% |
| The onset of tinnitus is related to | | |
| **Exposure to loud sounds** | 150 | 28.2% |
| **Change in hearing** | 27 | 5.1% |
| **Exposure to changes in surrounding pressure (such as flying or driving)** | 54 | 10.2% |
| **Flu, common cold, or other infections** | 115 | 21.6% |
| **Feeling of fullness or pressure in the ears** | 53 | 10% |
| **Anxiety** | 109 | 20.9% |
| **Head injury** | 28 | 5.3% |
| **Neck injury** | 9 | 1.7% |
| **Other; Ear infection** | 2 | 0.4% |
| **Other; I don't know** | 4 | 0.8% |
| **None circumstances** | 200 | 37.6% |
| Respondent drug intake | | |
| **Aspirin** | 10 | 1.9% |
| **Pain relievers** | 89 | 16.7% |
| **Antibiotics** | 49 | 9.2% |
| **Quinine** | 10 | 1.9% |
| **Diuretics** | 3 | 0.6% |
| **Antidepressants** | 7 | 1.3% |
| **None** | 315 | 59.2 |
| **I don't know** | 88 | 16.5 |

## Tinnitus-associated factors

**Table 6** presents the factors potentially associated with tinnitus. The Chi-square and Fisher's exact tests revealed that several factors were significantly associated with tinnitus (p-value < 0.05). These factors included demographic variables such as age and gender, as well as a family history of tinnitus, specifically in the mother, brothers, or participants who reported "I don't know" or "None." Significant associations were also found with symptoms like dizziness and various ear-related conditions, including acoustic trauma from sudden loud noise, middle ear infections due to external pressure, sudden hearing loss, middle ear infections or Eustachian tube dysfunction, and even the absence of any ear-related conditions. Medical history, such as previous dental procedures or the absence of any procedures, was also significant. Other associated variables included sensitivity to external sounds, difficulty hearing in noisy environments, and a range of symptoms such as headache, neck pain, ear pain, jaw pain, facial pain, or the absence of pain. Additionally, oral health issues like Jaw joint

**Table 5. Management of tinnitus (n = 532).**

| Variable | Frequency | Percentage(%) |
|---|---|---|
| Over the past year, have you seen your family doctor or seen a healthcare professional at a clinic or hospital about your tinnitus? | | |
| **Yes, 5 or more visits** | 8 | 1.5% |
| **Yes, 2–4 visits** | 18 | 3.4% |
| **Yes, only one visit** | 34 | 6.4% |
| **No, never** | 379 | 71.5% |
| **I don't know** | 93 | 17.5% |
| Tinnitus treatment | | |
| **Psychological therapy** | 5 | 0.9% |
| **Audiological therapy** | 5 | 0.9% |
| **Physical therapy** | 2 | 0.4% |
| **Self-management** | 16 | 3% |
| **None** | 504 | 94.7 |

**Table 6. Association between the presence of tinnitus with socio-demographic and clinical characteristics.**

| Characteristics | Tinnitus yes (n = 532) | Tinnitus no (*n* = 599) | p-Value (*Chi-Square Test/^ Fisher's Exact Test) |
|---|---|---|---|
| Age | | | **.028*** |
| **15** | 161 (14.2%) | 181 (16%) | |
| **16** | 151 (13.4%) | 213 (18.8%) | |
| **17** | 169 (14.9%) | 150 (13.3%) | |
| **18** | 51 (4.5%) | 55 (4.9%) | |
| Gender | | | **<.001*** |
| **Male** | 162 (40.4%) | 239 (59.6%) | |
| **Female** | 370 (50.7%) | 360 (49.3%) | |
| BMI | | | .244* |
| **Underweight >18.5** | 80 (7.1%) | 107 (9.5%) | |
| **Normal weight 18.524.9** | 356 (31.5%) | 390 (34.55) | |
| **Overweight 2529.9** | 70 (6.2%) | 83 (7.3%) | |
| **Obesity ≥ 30** | 26 (2.3%) | 18 (1.6%) | |
| City | | | .080* |
| **Hebron** | 205 (18.1%) | 217 (19.2%) | |
| **Nablus** | 119 (10.5%) | 166 (14.7%) | |
| **Ramallah** | 176 (15.6%) | 170 (15%) | |
| **Bethlehem** | 32 (2.8%) | 46 (4.1%) | |
| High Phone Usage | | | .172* |
| **Yes** | 345 (30.5%) | 356 (31.5%) | |
| **No** | 24 (2.1%) | 32 (2.8%) | |
| **Not much** | 163 (14.4%) | 211 (18.7%) | |
| Dominant Writing Hand | | | .154* |
| **Right** | 450 (39.8%) | 521 (46.1%) | |
| **Left** | 28 (2.5%) | 36 (3.2%) | |
| **Both** | 54 (4.8%) | 42 (3.7%) | |

*(Continued)*

**Table 6.** (Continued)

| Characteristics | Tinnitus yes (n = 532) | Tinnitus no (n = 599) | p-Value (*Chi-Square Test/^ Fisher's Exact Test) |
|---|---|---|---|
| First-Degree Relatives with Tinnitus | | | |
| **Mother** | 57 (5%) | 39 (3.4%) | **.011*** |
| **Father** | 33 (2.9%) | 24 (2.1%) | .092* |
| **Brothers** | 31 (2.7%) | 13 (1.1%) | **.002*** |
| **Sisters** | 26 (2.3%) | 19 (1.7%) | .141* |
| **I don't know** | 286 (25.3%) | 242 (21.4%) | **<.001*** |
| **None** | 154 (13.6%) | 303 (26.8%) | **<.001*** |
| Number of Affected Relatives | | | |
| **≤ 3** | 78 (54.5%) | 43 (30.1%) | |
| **4–6** | 12 (8.4%) | 9 (6.3%) | .761^ |
| **≥ 7** | 1 (0.7%) | 0 (0%) | |
| Dizziness | | | **<.001*** |
| **Yes, less than once a year** | 142 (12.6%) | 153 (13.5%) | |
| **Yes, more than once a year** | 288 (25.5%) | 195 (17.3%) | |
| **No** | 102 (9%) | 251 (22.2%) | |
| Ear-related conditions: | | | |
| Acoustic trauma due to sudden loud noise | | | **<.001*** |
| **Yes** | 31 (2.7%) | 10 (0.9%) | |
| **No** | 501 (44.3%) | 589 (52.1%) | |
| Middle ear infection due to external pressure | | | **<.001*** |
| **Yes** | 67 (5.9%) | 28 (2.5%) | |
| **No** | 465 (41.1%) | 571 (50.5%) | |
| Age-related hearing loss | | | .470^ |
| **Yes** | 1 (0.1%) | 0 (0%) | |
| **No** | 531 (46.9%) | 599 (53%) | |
| Sudden hearing loss | | | **.005^** |
| **Yes** | 7 (0.6%) | 0 (0%) | |
| **No** | 525 (46.4%) | 599 (53%) | |
| Acoustic neuroma | | | .470^ |
| **Yes** | 1 (0.1%) | 0 (0%) | |
| **No** | 531 (46.9%) | 599 (53%) | |
| Chronic ear infection | | | .093* |
| **Yes** | 17 (1.5%) | 10 (0.9%) | |
| **No** | 515 (45.5%) | 589 (52.1%) | |
| Otosclerosis | | | 1.000^ |
| **Yes** | 0 (0%) | 1 (0.1%) | |
| **No** | 532 (47%) | 598 (52.9%) | |
| Middle ear infection or Eustachian tube dysfunction | | | **.003*** |
| **Yes** | 19 (1.7%) | 6 (0.5%) | |
| **No** | 513 (45.5%) | 593 (52.4%) | |
| Eardrum perforation | | | .854* |
| **Yes** | 9 (0.85%) | 11 (1.0%) | |
| **No** | 523 (46.2%) | 588 (52%) | |
| Hearing loss due to other reasons | | | .327* |
| **Yes** | 10 (0.9%) | 7 (0.6%) | |
| **No** | 522 (46.2%) | 592 (52.3%) | |

*(Continued)*

**Table 6.** (Continued)

| Characteristics | Tinnitus yes (n = 532) | Tinnitus no (n = 599) | p-Value (*Chi-Square Test/^ Fisher's Exact Test) |
|---|---|---|---|
| None conditions | | | **<.001*** |
| Yes | 398 (35.2%) | 536 (47.4%) | |
| No | 134 (11.8%) | 63 (5.6%) | |
| Prior medical procedures: | | | |
| Ear surgery | | | .271* |
| Yes | 15 (1.3%) | 11 (1%) | |
| No | 517 (45.7%) | 588 (52%) | |
| Dental procedure (filling removal, dental implants, prolonged dental surgery) | | | |
| Yes | 132 (11.7%) | 96 (8.5%) | **<.001*** |
| No | 400 (35.4%) | 503 (44.5%) | |
| Neurosurgery | | | .292* |
| Yes | 8 (0.7%) | 5 (0.4%) | |
| No | 524 (46.3%) | 594 (52.5%) | |
| Spinal tap (lumbar puncture) | | | .204^ |
| Yes | 7 (0.6%) | 3 (0.3%) | |
| No | 525 (46.4%) | 596 (52.7%) | |
| Chemotherapy | | | 1.000^ |
| Yes | 2 (0.2%) | 3 (0.3%) | |
| No | 530 (46.9%) | 596 (52.7%) | |
| Radiation therapy for the head and neck | | | .916* |
| Yes | 5 (0.4%) | 6 (0.5%) | |
| No | 527 (46.6%) | 593 (52.4%) | |
| Electroconvulsive therapy | | | .671^ |
| Yes | 3 (0.3%) | 2 (0.2%) | |
| No | 529 (46.8%) | 597 (52.8%) | |
| Nasal polypectomy | | | 1.000^ |
| Yes | 2 (0.2%) | 2 (0.2%) | |
| No | 530 (46.9%) | 597 (52.8%) | |
| Tonsillectomy | | | .999* |
| Yes | 8 (0.7%) | 9 (0.8%) | |
| No | 524 (46.3%) | 590 (52.2%) | |
| Pulmonary Laceration | | | 1.000^ |
| Yes | 0 (0%) | 1 (0.1%) | |
| No | 532 (47%) | 598 (52.9%) | |
| Hand Fracture | | | .470^ |
| Yes | 1 (0.1%) | 0 (0%) | |
| No | 531 (46.9%) | 599 (53%) | |
| Sleeve Gastrectomy | | | .470^ |
| Yes | 1 (0.1%) | 0 (0%) | |
| No | 531 (46.9%) | 599 (53%) | |
| None procedures | | | **<.001*** |
| Yes | 368 (32.5%) | 478 (42.3%) | |
| No | 164 (14.5%) | 121 (10.7%) | |

*(Continued)*

**Table 6.** (Continued)

| Characteristics | Tinnitus yes (n = 532) | Tinnitus no (*n* = 599) | p-Value (*Chi-Square Test/^ Fisher's Exact Test) |
|---|---|---|---|
| Sensitivity to External Sounds during the past week | | | **<.001*** |
| **Yes** | 285 (25.2%) | 107 (9.5%) | |
| **No** | 247 (21.8%) | 492 (43.5%) | |
| Difficulty Hearing in Noisy Environments | | | **<.001*** |
| **No** | 207 (18.3%) | 412 (36.4%) | |
| **Slight difficulty** | 300 (24.5%) | 179 (15.8%) | |
| **Yes, I cannot hear at all** | 25 (2.2%) | 8 (0.7%) | |
| Hearing Assistive Device Usage: | | | |
| Hearing aid | | | .486^ |
| **Yes** | 5 (0.4%) | 3 (0.3%) | |
| **No** | 527 (46.6%) | 596 (52.7%) | |
| Cochlear implant | | | 1.000^ |
| **Yes** | 0 (0%) | 1 (0.1%) | |
| **No** | 532 (47%) | 598 (52.9%) | |
| Sound generator | | | .221^ |
| **Yes** | 2 (0.2%) | 0 (0%) | |
| **No** | 530 (46.9%) | 599 (53%) | |
| Combined device (hearing aid and sound generator in one device) | | | .604^ |
| **Yes** | 2 (0.2%) | 1 (0.1%) | |
| **No** | 530 (46.9%) | 598 (52.9%) | |
| None devices | | | .193* |
| **Yes** | 523 (46.2%) | 594 (52.5%) | |
| **No** | 9 (0.8%) | 5 (0.4%) | |
| Experience of Symptoms: | | | |
| Headache | | | **<.001*** |
| **Yes** | 301 (26.6%) | 173 (15.3%) | |
| **No** | 231 (20.4%) | 426 (37.7%) | |
| Neck pain | | | **<.001*** |
| **Yes** | 153 (13.5%) | 55 (4.9%) | |
| **No** | 379 (33.5%) | 544 (48.1%) | |
| Ear pain | | | **<.001*** |
| **Yes** | 88 (7.8%) | 24 (2.1%) | |
| **No** | 444 (39.3%) | 575 (50.8%) | |
| Jaw pain | | | **<.001*** |
| **Yes** | 74 (6.5%) | 27 (2.4%) | |
| **No** | 458 (40.5%) | 572 (50.6%) | |
| Facial pain | | | **<.001*** |
| **Yes** | 31 (2.7%) | 10 (0.9%) | |
| **No** | 501 (44.3%) | 589 (52.1%) | |
| Shoulder pain | | | .193^ |
| **Yes** | 4 (0.4%) | 1 (0.1%) | |
| **No** | 528 (46.7%) | 598 (52.9%) | |
| None Pain | | | **<.001*** |
| **Yes** | 173 (15.3%) | 400 (35.4%) | |
| **No** | 359 (31.7%) | 199 (17.6%) | |

*(Continued)*

| Characteristics | Tinnitus yes (n = 532) | Tinnitus no (n = 599) | p-Value (*Chi-Square Test/^ Fisher's Exact Test) |
|---|---|---|---|
| Oral conditions: | | | |
| Jaw joint pain | | | **<.001*** |
| Yes | 73 (6.5%) | 33 (2.9%) | |
| No | 459 (40.6%) | 566 (50%) | |
| Dental problems | | | **<.001*** |
| Yes | 211 (18.7%) | 146 (12.9%) | |
| No | 321 (28.4%) | 453 (40.1%) | |
| Neurological conditions: | | | |
| Meningitis | | | .268* |
| Yes | 7 (0.6%) | 4 (0.4%) | |
| No | 525(46.4%) | 595 (52.6%) | |
| Multiple sclerosis | | | **0.030^** |
| Yes | 7 (0.6%) | 1 (0.1%) | |
| No | 525 (46.4%) | 598 (52.9%) | |
| Epilepsy | | | **.018*** |
| Yes | 17 (1.5%) | 7 (0.6%) | |
| No | 515 (45.5%) | 592 (52.3%) | |
| Stroke | | | .604^ |
| Yes | 2 (0.2%) | 1 (0.1%) | |
| No | 530 (46.9%) | 598 (52.9%) | |
| Other cerebrovascular diseases | | | .092^ |
| Yes | 7 (0.6%) | 2 (0.2%) | |
| No | 525 (46.4%) | 597 (52.8%) | |
| Psychological conditions: | | | |
| Anxiety, excessive stress | | | **<.001*** |
| Yes | 264 (23.3%) | 162 (14.3%) | |
| No | 268 (23.7%) | 437 (38.6%) | |
| Depression | | | **<.001*** |
| Yes | 98 (8.7%) | 43 (3.8%) | |
| No | 434 (38.4%) | 556 (49.2%) | |
| Emotional trauma | | | **<.001*** |
| Yes | 59 (5.2%) | 23 (2%) | |
| No | 473 (41.8%) | 576 (50.9%) | |
| Sleep disorders: | | | |
| Difficulty falling asleep | | | **<.001*** |
| Yes | 246 (21.8%) | 126 (11.1%) | |
| No | 286 (25.3%) | 473 (41.8%) | |
| Difficulty staying asleep | | | **<.001*** |
| Yes | 122 (10.8%) | 58 (5.1%) | |
| No | 410 (36.6%) | 541 (47.8%) | |
| Cardiovascular conditions: | | | |
| Low blood pressure | | | .199* |
| Yes | 30 (2.7%) | 24 (2.1%) | |
| No | 502 (44.4%) | 575 (50.8%) | |

*(Continued)*

**Table 6.** (Continued)

| Characteristics | Tinnitus yes (n = 532) | Tinnitus no (*n* = 599) | p-Value (*Chi-Square Test/^ Fisher's Exact Test) |
|---|---|---|---|
| High blood pressure | | | **<.001*** |
| Yes | 29 (2.6%) | 6 (0.5%) | |
| No | 503 (44.5%) | 593 (52.4%) | |
| Myocardial infarction (heart attack) | | | 1.000^ |
| Yes | 1 (0.1%) | 1 (0.1%) | |
| No | 531 (46.9%) | 598 (52.9%) | |
| Endocrine and metabolic conditions: | | | |
| Thyroid dysfunction | | | **0.018*** |
| Yes | 17 (1.5%) | 7 (0.6%) | |
| No | 515 (45.5%) | 592 (52.3%) | |
| Diabetes | | | .529^ |
| Yes | 6 (0.5%) | 4 (0.4%) | |
| No | 526 (46.5%) | 595 (52.6%) | |
| High cholesterol | | | .020* |
| Yes | 9 (0.8%) | 2 (0.2%) | |
| No | 523 (46.2%) | 597 (52.85) | |
| Rheumatic and autoimmune disorders: | | | |
| Rheumatoid arthritis | | | **<.001*** |
| Yes | 18 (1.6%) | 4 (0.4%) | |
| No | 514 (45.4%) | 595 (52.6%) | |
| Lupus (facial rash) | | | .092^ |
| Yes | 7 (0.6%) | 2 (0.2%) | |
| No | 525 (46.4%) | 597 (52.8%) | |
| Ear, nose, and throat conditions: | | | |
| Chronic sinusitis | | | **<.001*** |
| Yes | 77 (6.8%) | 40 (3.5%) | |
| No | 455 (40.2%) | 559 (49.4%) | |
| Deviated nasal septum | | | .012* |
| Yes | 30 (2.7%) | 16 (1.4%) | |
| No | 502 (44.4%) | 583 (51.5%) | |
| Other conditions: | | | |
| Anemia | | | .029* |
| Yes | 47 (4.2%) | 33 (2.9%) | |
| No | 485 (42.9%) | 566 (50%) | |
| Balance disorder | | | **<.001*** |
| Yes | 55 (4.9%) | 24 (2.1%) | |
| No | 477 (42.2%) | 575 (50.8%) | |
| Gastroesophageal reflux disease (GERD) | | | .039* |
| Yes | 18 (1.6%) | 9 (0.8%) | |
| No | 514 (45.4%) | 590 (52.2%) | |
| Adenoid | | | .501^ |
| Yes | 0 (0%) | 2 (0.2%) | |
| No | 532 (47%) | 597 (52.8%) | |

*(Continued)*

**Table 6.** (Continued)

| Characteristics | Tinnitus yes (n = 532) | Tinnitus no (*n* = 599) | p-Value (*Chi-Square Test/^ Fisher's Exact Test) |
|---|---|---|---|
| Asthma | | | 1.000^ |
| Yes | 1 (0.1%) | 2 (0.2%) | |
| No | 531 (46.9%) | 597 (52.8%) | |
| Epistaxis | | | 1.000^ |
| Yes | 1 (0.1%) | 1 (0.1%) | |
| No | 531 (46.9%) | 598 (52.9%) | |
| Familial Mediterranean Fever | | | .470^ |
| Yes | 1 (0.1%) | 0 (0%) | |
| No | 531 (46.9%) | 599 (53%) | |
| HSV | | | 1.000^ |
| Yes | 0 (0%) | 1 (0.1%) | |
| No | 532 (47%) | 598 (52.9%) | |
| Helicobacter pylori | | | 1.000^ |
| Yes | 0 (0%) | 1 (0.1%) | |
| No | 532 (47%) | 598 (52.9%) | |
| Iron Deficiency | | | .470^ |
| Yes | 1 (0.1%) | 0 (0%) | |
| No | 531 (46.9%) | 599 (53%) | |
| Leukemia | | | 1.000^ |
| Yes | 0 (0%) | 1 (0.1%) | |
| No | 532 (47%) | 598 (52.9%) | |
| Migraine | | | .671^ |
| Yes | 3 (0.3%) | 2 (0.2%) | |
| No | 529 (46.8%) | 597 (52.8%) | |
| OCD | | | 1.000^ |
| Yes | 0 (0%) | 1 (0.1%) | |
| No | 532 (47%) | 598 (52.9%) | |
| Schizophrenia | | | 1.000^ |
| Yes | 0 (0%) | 1 (0.1%) | |
| No | 532 (47%) | 598 (52.9%) | |
| Parkinson's disease | | | .470^ |
| Yes | 1 (0.1%) | 0 (0%) | |
| No | 531 (46.9%) | 599 (53%) | |

pain and dental problems, neurological conditions such as epilepsy and multiple sclerosis, and psychological conditions including anxiety or excessive stress, depression, and emotional trauma were all significantly linked to tinnitus. Sleep problems, particularly difficulty falling asleep and staying asleep, were also associated. Further medical conditions, such as high blood pressure, thyroid dysfunction, high cholesterol, rheumatoid arthritis, chronic sinusitis, deviated nasal septum, anemia, balance disorders, and gastroesophageal reflux disease (GERD), showed significant relationships with tinnitus as well.

## Binary logistic regression analysis of the associations between tinnitus and influencing factors

Predictors of tinnitus were determined by entering significant variables from the Chi-square and Fisher's exact analysis into a binary logistic regression model. The analysis revealed that individuals aged 17 were significantly more likely to

have tinnitus by 1.5 times compared to those aged 15 (p = 0.038, 95% CI: 1.023–2.157). Additionally, reporting first-degree relatives had tinnitus was significantly associated with an approximately threefold increased risk of tinnitus (p = 0.010, 95% CI: 1.277–6.044). Moreover, higher sensitivity to external sounds during the past week was significantly associated with tinnitus, with a 2.7 times increased likelihood compared to those who didn't report such sensitivity (p < 0.001, 95% CI: 1.971–3.711). Slight difficulty hearing in noisy environments was also significantly associated with tinnitus, increasing the likelihood by 1.7 times compared to those without such difficulty (p < 0.001, 95% CI: 1.261–2.314). Reporting pain symptoms was significantly associated with tinnitus, doubling the risk compared to those who didn't report experiencing pain (p = 0.033, 95% CI: 1.057–3.805). Lastly, difficulty falling asleep was associated with a 1.8-fold higher risk of tinnitus compared to those who didn't report such difficulty (p < 0.001, 95% CI: 1.270–2.462). Results are shown in **Table 7**.

**Table 7. Logistic regression analysis of the associations between tinnitus and influencing factors.**

| Variable (reference) | B | p-Value | OR | 95% CI | |
|---|---|---|---|---|---|
| | | | | Lower | Upper |
| Age: **16 (15)** | .019 | .917 | 1.019 | .709 | 1.465 |
| **Age:** 17 (15) | **.396** | .038 | **1.486** | **1.023** | **2.157** |
| Age: 18 (15) | .101 | .711 | 1.107 | .647 | 1.894 |
| **Gender:** Female (Male) | **.188** | **.249** | **1.206** | **.877** | **1.660** |
| First-Degree Relatives with Tinnitus: **Mother (no)** | −.664 | .128 | .515 | .219 | 1.210 |
| **First-Degree Relatives with Tinnitus:** Brothers (no) | **.575** | **.215** | **1.778** | **.717** | **4.411** |
| First-Degree Relatives with Tinnitus: **I don't know (yes)** | .353 | .369 | 1.424 | .658 | 3.079 |
| **First-Degree Relatives with Tinnitus:** None (yes) | **1.022** | **.010** | **2.778** | **1.277** | **6.044** |
| Dizziness: **Yes, less than once a year (no)** | .082 | .680 | 1.086 | .733 | 1.608 |
| **Dizziness:** Yes, more than once a year (no) | **.091** | **.647** | **1.096** | **.742** | **1.619** |
| Ear-related conditions: **Acoustic trauma due to sudden loud noise (no)** | .779 | .120 | 2.180 | .816 | 5.824 |
| Ear-related conditions: Middle ear infection due to external pressure (no) | **.676** | **.097** | **1.966** | **.884** | **4.370** |
| Ear-related conditions: **Sudden hearing loss (no)** | 19.779 | .999 | 389085857.96 | .000 | |
| **Ear-related conditions:** Middle ear infection or Eustachian tube dysfunction (no) | **.560** | **.359** | **1.751** | **.529** | **5.796** |
| Ear-related conditions: **None conditions (yes)** | −.133 | .704 | .875 | .440 | 1.741 |
| **Prior medical procedures:** Dental procedure (filling removal, dental implants, prolonged dental surgery) (no) | **.020** | **.957** | **1.020** | **.497** | **2.095** |
| Prior medical procedures: **None procedures (Yes)** | .162 | .632 | 1.176 | .606 | 2.282 |
| **Sensitivity to External Sounds during the past week** (no) | **.995** | **<.001** | **2.704** | **1.971** | **3.711** |
| Difficulty Hearing in Noisy Environments: **Slight difficulty (no)** | .536 | **<.001** | 1.708 | 1.261 | 2.314 |
| **Difficulty Hearing in Noisy Environments:** Yes, I cannot hear at all (no) | **.706** | **.144** | **2.025** | **.786** | **5.219** |
| Symptoms: **Headache (no)** | −.144 | .633 | .866 | .478 | 1.567 |
| Symptoms: Neck pain (no) | **.369** | **.113** | **1.447** | **.916** | **2.286** |
| Symptoms: **Ear pain (no)** | .292 | .322 | 1.340 | .751 | 2.388 |
| **Symptoms**: Jaw pain (no) | **.049** | **.884** | **1.050** | **.544** | **2.027** |
| Symptoms: **Facial pain (no)** | −.478 | .298 | .620 | .252 | 1.527 |
| **Symptoms:** None Pain (yes) | **.696** | **.033** | **2.006** | **1.057** | **3.805** |
| Oral conditions: **Jaw joint pain (no)** | −.104 | .741 | .902 | .488 | 1.665 |

*(Continued)*

| Variable (reference) | B | p-Value | OR | 95% CI | |
|---|---|---|---|---|---|
| | | | | Lower | Upper |
| **Oral conditions:** Dental problems (no) | .082 | .623 | 1.086 | .782 | 1.507 |
| Neurological conditions: **Multiple sclerosis (no)** | .645 | .609 | 1.906 | .161 | 22.566 |
| **Neurological conditions:** Epilepsy (no) | .169 | .771 | 1.185 | .378 | 3.714 |
| Psychological conditions: **Anxiety, excessive stress (no)** | .126 | .439 | 1.134 | .824 | 1.561 |
| **Psychological conditions:** Depression (no) | .073 | .765 | 1.076 | .665 | 1.742 |
| Psychological conditions: **Emotional trauma (no)** | .310 | .326 | 1.363 | .735 | 2.528 |
| **Sleep disorders:** Difficulty falling asleep (no) | .570 | <.001 | 1.768 | 1.270 | 2.462 |
| Sleep disorders: **Difficulty staying asleep (no)** | .209 | .327 | 1.233 | .812 | 1.873 |
| **Cardiovascular conditions:** High blood pressure (no) | .834 | .141 | 2.301 | .758 | 6.984 |
| Endocrine and metabolic conditions: **Thyroid dysfunction (no)** | .151 | .795 | 1.163 | .372 | 3.632 |
| **Endocrine and metabolic conditions:** High cholesterol (no) | .806 | .459 | 2.239 | .265 | 18.931 |
| Rheumatic and autoimmune disorders: **Rheumatoid arthritis (no)** | .813 | .228 | 2.255 | .601 | 8.459 |
| **Ear, nose, and throat conditions:** Chronic sinusitis (no) | -.073 | .785 | .930 | .551 | 1.568 |
| Ear, nose, and throat conditions: **Deviated nasal septum (no)** | -.283 | .495 | .754 | .334 | 1.698 |
| **Other conditions**: Anemia (no) | -.257 | .387 | .773 | .432 | 1.384 |
| **Other conditions: Balance disorder** | -.067 | .831 | .935 | .506 | 1.729 |
| **Other conditions**: Gastroesophageal reflux disease (GERD) (no) | -.240 | .637 | .787 | .290 | 2.132 |

### Binary logistic regression analysis of the associations between tinnitus subtypes and influencing factors

Within the group of participants with tinnitus (n = 532), the predictor variables were evaluated. The variables that were significant in the chi-square analysis (results not included) were introduced into the logistic regression analysis to determine the predictors for having objective tinnitus compared with the subjective subtype and intermittent tinnitus compared with the continuous subtype. Table 8 shows the variables that were significantly associated with having objective or subjective tinnitus in the first column; however, the variables that were significant predictors of having objective tinnitus based on the logistic regression were just having epilepsy and migraine. Participants with tinnitus who have epilepsy are more likely to have objective tinnitus by around 9 times (p = 0.001, 95% CI: 2.59–37.579). Additionally, participants with tinnitus who have migraine are more likely to have objective tinnitus by around 16 times (p = 0.032, 95% CI: 1.268–208.176). Table 9 shows the variables that were significantly associated with having intermittent or continuous tinnitus in the first column; however, the variables that were significant predictors for continuous tinnitus were prior tonsillectomy and sensitivity to external sounds. Participants with tinnitus who have had tonsillectomy are less likely to have continuous tinnitus (OR: 0.164), (p = 0.03, 95% CI: 0.032–0.836). Additionally, participants with tinnitus who were sensitive to external sounds are also less likely to have continuous tinnitus (OR: 0.0619), (p = 0.013, 95% CI: 0.424–0.903).

### Discussion

The reported tinnitus prevalence in the current study is 47% among Palestinian young people aged 15–18 years old, which is notably high. The significant association between age and tinnitus in the current study, with older teenager

**Table 8. Logistic regression analysis of the associations between tinnitus form (subjective vs. objective) and the patients' characteristics.**

| Variable (reference) | B | p-Value | OR | 95% CI | |
|---|---|---|---|---|---|
| | | | | Lower | Upper |
| First-Degree Relatives with Tinnitus: Mother (no) | .397 | .559 | 1.487 | .392 | 5.637 |
| | .492 | .526 | 1.636 | .358 | 7.472 |
| First-Degree Relatives with Tinnitus: Sister (no) | .002 | .999 | 1.002 | .179 | 5.604 |
| | −.293 | .554 | .746 | .283 | 1.970 |
| Ear-related conditions: Acoustic trauma due to sudden loud noise (no) | .405 | .577 | 1.500 | .361 | 6.237 |
| Ear-related conditions: Sudden hearing loss (no) | 1.217 | .273 | 3.378 | .383 | 29.786 |
| Ear-related conditions: Eardrum perforation (no) | −.034 | .975 | .966 | .113 | 8.243 |
| Ear-related conditions: None conditions (yes) | −.963 | .062 | .382 | .139 | 1.051 |
| Prior medical procedures: Radiation therapy for the head and neck (no) | 1.557 | .211 | 4.743 | .413 | 54.440 |
| Sensitivity to External Sounds during the past week (No) | −.094 | .834 | .910 | .379 | 2.184 |
| Hearing Assistive Device Usage: Sound Generator (no) | 2.390 | .254 | 10.911 | .179 | 664.963 |
| Hearing Assistive Device Usage: None (yes) | −.662 | .605 | .516 | .042 | 6.345 |
| Neurological conditions: Meningitis (no) | 1.426 | .183 | 4.162 | .511 | 33.918 |
| Neurological conditions: Epilepsy (no) | 2.289 | **0.001** | 9.865 | 2.590 | 37.579 |
| Cardiovascular conditions: High BP (no) | .308 | .696 | 1.360 | .291 | 6.366 |
| Endocrine and metabolic conditions: Thyroid Dysfunction (no) | .702 | .422 | 2.017 | .364 | 11.181 |
| Endocrine and metabolic conditions: Diabetes (no) | .760 | .562 | 2.138 | .164 | 27.932 |
| Other conditions: Epistaxis (no) | 22.499 | 1.000 | 5907292947.117 | 0.000 | |
| Other conditions: Migraine (no) | 2.788 | **0.032** | 16.249 | 1.268 | 208.176 |

**Table 9. Logistic regression analysis of the associations between tinnitus form (intermittent vs. continuous) and the patients' characteristics.**

| Variable (reference) | B | p-value | OR | 95% CI | |
|---|---|---|---|---|---|
| | | | | Lower | Upper |
| Prior medical procedures: Tonsillectomy (no) | −1.808 | **0.030** | .164 | .032 | .836 |
| Sensitivity to External Sounds during the past week (No) | −.480 | **0.013** | .619 | .424 | .903 |
| Hearing Assistive Device Usage: Combined Device(no) | −21.759 | 0.999 | .000 | 0.000 | – |

(17-year-olds) exhibiting a 1.5-fold increased risk compared to younger studied group (15-year-olds). This is supported by previous studies that showed the prevalence of tinnitus increases with age [1,26,27] and among adolescents, those in their mid-teens have the highest incidence of tinnitus [26]. The single-year ages of 15 and 17 years examined in the present study both fall unequivocally within middle adolescence from a developmental perspective [20]. Their comparison should not be viewed as representing discrete developmental stages. The use of age ranges in adolescent research is theoretically based on the understanding that development during this life stage is continuous, non-linear, and not limited by rigid chronological limits, acknowledging contemporary perspectives that emphasize the fluidity and extended upper boundary of adolescence [28]. Adolescence is still a relatively understudied and developing life stage, with growing understanding that its upper boundary goes beyond traditional bounds.

Gender differences, more prevalent among females, align with prior research findings [29–31]. This may be due to the greater tendency of girls to describe symptoms [32] and their more frequent generation of spontaneous otoacoustic emissions [33]. Our results revealed that tinnitus was more prevalent in females (50.7%) than in males (40.4%). Increased prevalence of tinnitus and tinnitus annoyance in females was also shown in a survey on the population of South Korea by Park et al. Those authors attributed their findings to the more stressful cultural situation of South Korean women, which

demands female obedience and more roles for women within the family [34]. Palestinian females suffer similar cultural stressors in addition to political violence arising from the military occupation [35], which may have contributed to the increased perception of tinnitus in girls in our study. On the other hand, the decreased prevalence of tinnitus in Palestinian males may arise from the tendency of males to not declare their health problems [36].

Notably, the current study indicated that a positive family history of tinnitus increases the risk of developing tinnitus. This finding is in line with a previous study on tinnitus among Palestinian university students [18]. These notes support recent hypotheses about the possible genetic component in the etiology of tinnitus [37]. A recent genome-wide association study highlighted several significant pathways implicated in tinnitus [38]. Another study on tinnitus found a higher concordance rate in monozygotic twins compared to dizygotic twins [39]. Nevertheless, other large-scale studies have reported low heritability estimates for tinnitus and concluded that a strong association with any specific genetic locus is lacking [40]. It is frequently challenging to distinguish whether familial aggregation is caused by common genes or shared environmental factors, even though it has been demonstrated for many, if not all, disorders [41]. The existence of a familial impact for tinnitus allows for targeted research to ascertain if genetic factors or a shared familial environment are responsible for this effect [42]. Families are frequently exposed to the same environmental risks, eat comparable meals, and reside in the same geographic area. These common exposures may raise the chance of contracting specific illnesses, which could result in familial aggregation [43]. Because families live near one another and share living quarters, infectious diseases can spread readily within them. This may cause several family members to contract the same virus, giving the impression that there is a hereditary connection when in fact it is the result of transmission [44]. Family dynamics, stress levels, and learned coping mechanisms can also play a role. For example, a family with high levels of chronic stress might experience a higher prevalence of stress-related illnesses, even without a genetic predisposition [43]. Further collaborative attempts are necessary to provide biological insight into the potential genetic etiology of tinnitus.

Tinnitus in our sample frequently co-occurred with symptoms such as dizziness, sound sensitivity, headaches, and sleep disturbances, reflecting the multifaceted nature of the condition. These findings are consistent with previous research indicating that tinnitus often co-occurs with other auditory and non-auditory symptoms, including hyperacusis and psychological distress [45]. This study found a substantial association between disturbances in sleeping, especially difficulty in initiating sleep, and tinnitus. This corresponds with meta-analyses indicating that more than fifty percent of tinnitus sufferers suffer from sleep disturbances. The reciprocal association between tinnitus and sleep disruptions indicates that therapies aimed at improving sleep quality may positively impact tinnitus severity and vice versa [46].

According to the data, those affected experienced varying frequencies of tinnitus. Participants had tinnitus on a monthly, weekly, and daily basis with 13.5%, 17.9%, and 12.6% respectively, while a major proportion of participants experienced tinnitus every few months (28.2%) and yearly (27.8%). The frequency of tinnitus differs among individuals, suggesting it is a symptom rather than a disease, with multiple potential etiologies contributing to its onset. The figures indicate that a significant percentage of individuals experience tinnitus monthly, weekly, or daily. This data aligns with the evolving characteristics of tinnitus, emphasizing the necessity for a more profound understanding of its impact on individuals' lives [47]. The other characteristics of tinnitus in our study sample confirm the disorder's variability, with 32.7% reporting constant tinnitus and 67.3% experiencing intermittent tinnitus. The most prevalent tinnitus noises are crickets at 34.8%, noise at 25.7%, and tone sounds at 27.6%. Greater variation exists in the tinnitus rhythm, the affected ear, and whether it is subjective or objective. Notably, tinnitus is reported bilaterally in approximately 41% of our study population, indicating a potential association with systemic rather than localized disorders, such as Ménière's disease, vestibular schwannoma, atmospheric inner ear barotrauma, vertebrobasilar ischemic stroke, and otosclerosis [48,49]. Although the precise etiology of tinnitus remains unclear, numerous risk factors have been identified. Multiple environmental and health-related factors were identified as associated with tinnitus. Older age, female, history of smoking, sleep disturbances, stress, exposure to noise, and a history of various medical conditions such as arthritis, asthma, and thyroid disorders have been identified as risk factors for tinnitus [9,31].

A significant proportion of respondents (57.7%) reported uncertainty about the onset of their tinnitus, indicating a potential lack of awareness or difficulty in pinpointing the exact timing of symptoms. The high proportion of uncertain responses regarding tinnitus onset reflects findings from Rosing et al. (2016), who noted similar challenges in pediatric tinnitus case histories. Among those who could recall, the majority (18.4%) experienced tinnitus within the past 1–12 months, followed by 13.5% reporting onset within 13–24 months. This suggests that tinnitus may often develop relatively recently in this population, though a subset (2.6%) reported symptoms persisting for over 36 months. The high percentage of uncertain responses underscores the need for improved education and awareness about tinnitus in adolescents [29]. The timing of tinnitus-associated conditions relative to tinnitus onset revealed that only a small fraction of respondents (2.6%) experienced these before tinnitus began, while 4.3% reported conditions arising after tinnitus started. A negligible proportion (1.1%) noted coincident timing. The low percentages suggest that while some conditions may be linked to tinnitus, the majority of cases (92.8%) either had no clear association or were unknown. This highlights the complexity of identifying direct causative factors and the need for further research into underlying mechanisms.

Regarding medication use, the majority of respondents (59.2%) reported no drug intake, which contrasts with adult populations [50], suggesting different risk profiles in adolescents. The pain relievers use (16.7%) could imply a potential association with over-the-counter medications [51]. Antibiotics were reported by 9.2% of respondents, raising questions about ototoxic effects in some cases. However, the high percentage of "I don't know" responses (16.5%) indicates a need for better documentation and awareness of medication histories in this population. These findings emphasize the multifactorial nature of tinnitus in adolescents, with noise exposure, infections, and anxiety emerging as prominent risk factors [6]. The high prevalence of uncertainty regarding onset and associated conditions highlights the challenges in diagnosing and managing tinnitus in this age group. Public health efforts should focus on preventive measures, such as hearing protection and awareness campaigns, while clinicians should consider comprehensive evaluations that include mental health and medication histories [52].

The finding that 71.5% of participants had never sought professional help for tinnitus underscores a widespread reluctance or barrier to care. A recent scoping review reported that across diverse populations, fewer than one-third of individuals with tinnitus pursue medical evaluation, often due to beliefs that nothing can be done or low perceived severity [53]. Likewise, in Saudi Arabia, despite a prevalence of 37.6%, only a minority sought medical support, mirroring our finding of low help-seeking behavior [54].

Inner-ear pathologies such as acoustic trauma, middle-ear infections, and sudden hearing loss were significantly linked to tinnitus in our cohort, consistent with reports that cochlear injury and Eustachian tube dysfunction predispose to aberrant neural firing underlying phantom sound perception. Dizziness and balance disorders often coexist with tinnitus, reflecting shared vestibular–auditory pathophysiology [55]. Tinnitus frequently co-occurs with psychiatric conditions: anxiety, depression, and post-traumatic stress have been shown to exacerbate tinnitus distress and may influence its onset [56]. Sleep disturbances, particularly insomnia, both worsen symptom perception and are a consequence of tinnitus-related arousal. Pain syndromes—including headache, neck pain, and temporomandibular disorders—share convergent neural circuits with tinnitus, potentially explaining the association we observed [57]. Importantly, these findings highlight the necessity of adopting a biopsychosocial framework when interpreting tinnitus in adolescents [19]. In our Palestinian sample, the high prevalence of anxiety, stress, and sleep disturbances may reflect not only individual predisposition but also broader contextual stressors, including ongoing political conflict [58], academic pressures [22], and restricted access to healthcare [58], that can amplify tinnitus perception and distress. This underscores that tinnitus in this population cannot be understood through biological pathways alone but must account for the psychosocial environment in which these adolescents develop. Moreover, while tinnitus emerged as a reported symptom, it should be understood within the broader context of trauma-related conditions such as PTSD, particularly in conflict-affected areas such as Palestine, where the population, including all age groups, is frequently experiencing collective violence and adverse childhood experiences (ACEs) [59].

In our multivariate model, six factors emerged as independent predictors of tinnitus: adolescents aged 17 showed a 1.5-fold higher risk compared to those aged 15 (p = 0.038, 95% CI: 1.023–2.157), consistent with documented increases in tinnitus prevalence during late adolescence and early adulthood [60]; having a first-degree family history of tinnitus conferred approximately a threefold elevated risk (p = 0.010, 95% CI: 1.277–6.044), underscoring genetic predisposition evidenced by familial aggregation and heritability studies [61,62]; recent sensitivity to external sounds—hyperacusis—was associated with a 2.7-times greater likelihood of tinnitus (p < 0.001, 95% CI: 1.971–3.711), in line with the tight comorbidity observed between these conditions [63]; slight difficulty hearing in noisy environments increased the odds by 1.7 times (p < 0.001, 95% CI: 1.261–2.314), reflecting early sensorineural deficits as precursors to tinnitus [64]; reporting pain symptoms doubled the risk of tinnitus (p = 0.033, 95% CI: 1.057–3.805), corroborating population-based associations between chronic pain and tinnitus [65]; and difficulty falling asleep was linked to a 1.8-fold increased risk (p < 0.001, 95% CI: 1.270–2.462), consistent with evidence that insomnia both predisposes to and exacerbates tinnitus distress [66].

## Conclusions

This study demonstrates a remarkably high prevalence of tinnitus among young Palestinians aged 15–18. The findings suggest that tinnitus onset and severity are influenced by a number of risk factors, particularly age, family history, sleep disturbances, and comorbid symptoms such as hyperacusis and pain. A significant association between tinnitus and positive family history of tinnitus was noted, supporting the hypothesis of a genetic role. These findings underscore the need for targeted public health initiatives to raise awareness, promote early identification, and encourage appropriate management of tinnitus among adolescents.

### Recommendations

Future research should aim to clarify causal relationships and further explore the genetic, environmental, and psychosocial contributors to tinnitus in young populations. In particular, subtype-specific risk analysis, distinguishing between continuous vs. intermittent, subjective vs. objective tinnitus, and tinnitus with or without ear-related comorbidities, will be crucial for advancing knowledge and clinical practice. Such analyses will require larger or clinically enriched samples to enable sufficient statistical power and more targeted prevention and intervention strategies. Additionally, future studies should collect medication use data from all participants, regardless of tinnitus status, to enable a more robust assessment of the potential role of analgesics in the onset and severity of tinnitus. Future studies should incorporate comprehensive audiological and otologic evaluations to improve diagnostic accuracy and allow for clearer differentiation of tinnitus subtypes. Collaboration with audiologists and otolaryngologists is recommended to provide a more clinically grounded understanding of tinnitus and its underlying risk factors.

### Limitations

This study has several limitations to be considered when interpreting the results. First, its cross-sectional design limits our ability to establish causal relationships, as data were collected at a single point in time. Second, the reliance on self-reported measures may introduce information bias due to recall inaccuracies or subjective interpretations. Third, while the study included a large sample of Palestinian adolescents, the use of convenience sampling, necessitated by Ministry of Education restrictions and mobility barriers during the ongoing conflict, may introduce selection bias that could affect prevalence estimates. The non-probability sampling design does not allow formal estimation of the magnitude or direction of this bias; therefore, the findings should be interpreted with caution and cannot be generalized to the broader Palestinian population. Additionally, the absence of objective audiological assessments and otologic evaluations limits the ability to clinically verify tinnitus and differentiate between tinnitus subtypes. The prevalence of tinnitus might be overestimated in the current study, as individuals with pre-existing auditory problems may have been more motivated to participate. Individuals with access to digital platforms were able to participate, which represents an additional bias in participation. Since

identity confirmation was not applicable in our study setting, the precision of age-specific findings requires future studies with stricter verification protocols.

## Supporting information

**S1 File. Questionnaire.**
(DOCX)

**S2 File. SPSS Data Modified.**
(PDF)

## Acknowledgments

We thank everyone who contributed their time to make this project a reality. Likewise, we are grateful to the faculty of medicine and health sciences of An-Najah National University for their affordable collaboration.

## Author contributions

**Conceptualization:** Saad Al-Lahhaam, Aman Maraqa, Tala Albadawi, Raghad Doufish, Wa'd Amer, Mustafa Ghanim, Mohammad Abuawad, Maha Rabayaa, Malik ALQUB.

**Data curation:** Saad Al-Lahhaam, Malik ALQUB.

**Formal analysis:** Saad Al-Lahhaam, Raghad Dweikat, Tala Nazzal, Joud Khalil, Tala Albadawi, Raghad Doufish, Wa'd Amer, Mustafa Ghanim, Mohammad Abuawad, Amer Ghrouz, Samar Alkhaldi, Malik ALQUB.

**Investigation:** Saad Al-Lahhaam, Raghad Dweikat, Aman Maraqa, Raghad Doufish, Wa'd Amer, Mustafa Ghanim, Samar Alkhaldi, Maha Rabayaa, Malik ALQUB.

**Methodology:** Raghad Dweikat, Aman Maraqa, Joud Khalil, Tala Albadawi, Raghad Doufish, Wa'd Amer, Mustafa Ghanim, Samar Alkhaldi, Majdi Dwikat, Maha Rabayaa, Malik ALQUB.

**Software:** Tala Nazzal, Amer Ghrouz, Majdi Dwikat, Maha Rabayaa, Malik ALQUB.

**Supervision:** Saad Al-Lahhaam, Majdi Dwikat.

**Validation:** Saad Al-Lahhaam, Tala Nazzal, Tala Albadawi, Mohammad Abuawad, Amer Ghrouz, Samar Alkhaldi, Laith El-lahham, Maha Rabayaa, Malik ALQUB.

**Visualization:** Raghad Dweikat, Tala Nazzal, Aman Maraqa, Joud Khalil, Tala Albadawi, Raghad Doufish, Wa'd Amer, Mustafa Ghanim, Mohammad Abuawad, Amer Ghrouz, Samar Alkhaldi, Majdi Dwikat, Laith El-lahham, Maha Rabayaa, Malik ALQUB.

**Writing – original draft:** Saad Al-Lahhaam, Raghad Dweikat, Tala Nazzal, Aman Maraqa, Joud Khalil, Tala Albadawi, Raghad Doufish, Wa'd Amer, Mustafa Ghanim, Mohammad Abuawad, Amer Ghrouz, Samar Alkhaldi, Majdi Dwikat, Laith El-lahham, Maha Rabayaa, Malik ALQUB.

**Writing – review & editing:** Raghad Dweikat, Tala Nazzal, Aman Maraqa, Joud Khalil, Tala Albadawi, Raghad Doufish, Wa'd Amer, Mustafa Ghanim, Mohammad Abuawad, Amer Ghrouz, Samar Alkhaldi, Majdi Dwikat, Laith El-lahham, Maha Rabayaa.

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
