## [Decision Letter · Decision Letter 0]

3 Dec 2025

Dear Dr. ALQUB,

Thank you for submitting your manuscript to PLOS ONE. After careful consideration, we feel that it has merit but does not fully meet PLOS ONE’s publication criteria as it currently stands. Therefore, we invite you to submit a revised version of the manuscript that addresses the points raised during the review process.

We look forward to receiving your revised manuscript.

Kind regards,

Paul H Delano, Ph.D.

Academic Editor

PLOS ONE

Additional Editor Comments:

Please add some discussion about the actual context of Palestine and how it might increase anxiety, insomnia and tinnitus

Reviewers' comments:

Reviewer's Responses to Questions

**Comments to the Author**

1. Is the manuscript technically sound, and do the data support the conclusions?

Reviewer #1: Yes

Reviewer #2: Partly

Reviewer #3: Yes

2. Has the statistical analysis been performed appropriately and rigorously?

Reviewer #1: Yes

Reviewer #2: No

Reviewer #3: Yes

3. Have the authors made all data underlying the findings in their manuscript fully available?

Reviewer #1: Yes

Reviewer #2: No

Reviewer #3: Yes

4. Is the manuscript presented in an intelligible fashion and written in standard English?

Reviewer #1: Yes

Reviewer #2: Yes

Reviewer #3: Yes

Reviewer #1: Thank you very much for the opportunity to review this article. I find it highly interesting, and its findings undoubtedly warrant attention from public health policymakers.

The assumption of a 50% prevalence for the sample size calculation (which is striking because the prevalence in adults is 30%) is well justified because there were no previous regional data for the specific target population. It is understood that they assumed that tinnitus may have more pronounced risk factors in young people (e.g., exposure to loud music, academic stress levels, etc.). Without specific data for this age group, the researchers took the most rigorous and conservative route from a statistical point of view.

The only significant methodological weakness of this study is the use of convenience sampling, which introduces a possible selection bias and prevents the results from being generalised to the entire adolescent population of Palestine. However, this limitation is perfectly managed. The authors justify it due to the difficult conditions for research in the region, openly declare it, and explain its consequences.

The ‘Discussion’ section is particularly robust. The authors not only report their findings, but also contextualise them within the global scientific literature, discussing both the evidence that supports them and that which contradicts them.

The study is a very good example of rigorous and ethical research in an environment with logistical constraints.

Reviewer #2: Thank you for the opportunity to review this manuscript. It represents a commendable effort, and the article is generally well written and clearly communicated. I found it an engaging read. I do, however, have several important observations that I believe the authors should address to further strengthen the work.

Most of the concerns regarding this article stem from the limited information provided to evaluate the extent of bias introduced, particularly given the impossibility of randomization in the sample selection. This last consideration is a common issue in some attempts to establish population metrics, and it is well justified in this case in the text.

Although the final sample size is more than double the initially estimated target, the risk of substantial bias remains considerable. The authors acknowledge several potential sources of bias in their limitations section; however, they do not provide sufficient data to assess the magnitude of these issues, among others.

• Participation bias. The sampling strategy appears to favor individuals with auditory problems, which could significantly distort the representativeness of the findings. This potential bias is not discussed when comparing the results with other studies that may have relied on more representative sampling strategies.

• Access-related participation bias. A greater likelihood of participation among individuals with easier access to digital platforms introduces another layer of bias, potentially confounding, in an epidemiological fashion, the reported associations and limiting the external validity of the findings.

• Lack of quality control. No procedures are described to verify the identity or characteristics of respondents. While this is a recognized challenge in open surveys, it should be explicitly addressed, particularly given that the study population is narrowly defined in terms of age.

• Reflection about the instrument. Although I was not previously familiar with this instrument, it appears to have been widely applied across different settings. However, several methodological aspects require clarification. Was the instrument reverse-translated to ensure linguistic accuracy? Was a pilot study conducted prior to full implementation? In addition, given the length of the questionnaire, how did the authors account for the potential bias arising from respondent fatigue, whereby answers toward the end may be less accurate or consistent?

• Survey outreach. An estimation of the total population reached through different recruitment strategies would be valuable for evaluating coverage and representativeness.

• Attrition and completion rates. The manuscript does not report the number of participants who completed the survey relative to the number of attempts. It is implausible that all individuals who began the survey completed it. Similarly, it is unclear whether all data fields were fully completed, or whether the survey platform required mandatory responses to proceed. Clarifying this aspect would be essential, as attrition rates and missing data patterns are key indicators of data quality and potential bias.

In summary, while the sample size achieved is commendable, the absence of detailed information on recruitment, response validation, and attrition undermines confidence in the robustness and representativeness of the findings.

The presentation of results is excessively detailed, which diminishes the reader’s ability to discern the specific contributions of the article. Given that the topic is described as multifactorial and that the supporting evidence for associations with diverse risk factors remains generally weak, the simultaneous testing of a large number of variables is problematic. Moreover, some analyses based on the absence of associations within the dataset are presented but not adequately developed in the discussion. I recommend that the authors streamline their results by emphasizing those factors that appear most compelling or most closely aligned with the evidence introduced in the background, thereby strengthening the coherence and contribution of the manuscript.

Related to the previous point and regarding to the limited robustness of the sample, the use of modeling strategies may be problematic. The article would benefit from adopting a more explicitly exploratory tone, consistently avoiding the implication that the findings are representative of the entire population. For example, the results should be systematically framed as applying in “this sample of young Palestinians”, or something similar, rather than presented as generalizable estimates. In this light, the modeling exercises can still be considered valuable, but their contribution should be interpreted as illustrative and exploratory, rather than as evidence of definitive or population-level veracity.

There appears to be a systematic error in how prevalence percentages are calculated and communicated. For example, the second paragraph of the Discussion states that girls have a higher prevalence than boys, yet the percentages reported (32.7% and 14.3%) are inconsistent with the tabulated data. Using the authors’ own figures—730 girls in total (Table 1), of whom 370 have tinnitus (Table 6)—the prevalence among girls should be 370/730 ≈ 50.7%, not 32.7%. This suggests the denominator used was the total sample rather than the sex-specific subgroup. Please carefully review all other variables to ensure this error is not repeated, and re-evaluate whether comparisons with external studies remain valid in light of the corrected estimates.

Reviewer #3: The partial consideration of tinnitus as a psychosocial symptom is striking, given the context, both in the introduction and the discussion. The allusion to the healthcare costs of the problem is taken out of context and is not revisited in the discussion; it might be better to remove it.

**Do you want your identity to be public for this peer review?** For information about this choice, including consent withdrawal, please see our Privacy Policy

Reviewer #1: **Yes:** Lorena Rodríguez-Osiac

Reviewer #2: No

Reviewer #3: No

---

## [Author Response · Author response to Decision Letter 1]

21 Jan 2026

Dear editor,

Thank you for your time and we are grateful to the respected reviewers for their valuable comments which we believe they enhanced the manuscript. Kindly find below our responses. We highlighted the responses in the revised manuscript in red color.

Kind regards,

Malil Alqub, corresponding author, on behalf of all authors

Reviewer #1: Thank you very much for the opportunity to review this article. I find it highly interesting, and its findings undoubtedly warrant attention from public health policymakers.

We sincerely thank the reviewer for this positive assessment of the study’s relevance.

The assumption of a 50% prevalence for the sample size calculation (which is striking because the prevalence in adults is 30%) is well justified because there were no previous regional data for the specific target population. It is understood that they assumed that tinnitus may have more pronounced risk factors in young people (e.g., exposure to loud music, academic stress levels, etc.). Without specific data for this age group, the researchers took the most rigorous and conservative route from a statistical point of view.

We thank the reviewer for acknowledging and clearly articulating the rationale behind our sample size calculation. As noted, in the absence of prior regional data for Palestinian adolescents, adopting a 50% prevalence estimate was intended to ensure a conservative and statistically rigorous approach.

The only significant methodological weakness of this study is the use of convenience sampling, which introduces a possible selection bias and prevents the results from being generalised to the entire adolescent population of Palestine. However, this limitation is perfectly managed. The authors justify it due to the difficult conditions for research in the region, openly declare it, and explain its consequences.

We appreciate the reviewer’s balanced evaluation of this limitation. We have made every effort to transparently report the use of convenience sampling, justify it within the challenging research context, and clearly state its implications for generalisability in the manuscript.

The ‘Discussion’ section is particularly robust. The authors not only report their findings, but also contextualise them within the global scientific literature, discussing both the evidence that supports them and that which contradicts them.

We are grateful for this encouraging feedback. Considerable effort was devoted to ensuring that the discussion provided a critical and balanced synthesis of the findings within the broader international literature.

The study is a very good example of rigorous and ethical research in an environment with logistical constraints.

We sincerely thank the reviewer for this thoughtful and affirming comment. Conducting rigorous and ethical research under challenging conditions was a central priority of this study, and we greatly appreciate the reviewer’s recognition of this effort.

Reviewer #2: Thank you for the opportunity to review this manuscript. It represents a commendable effort, and the article is generally well written and clearly communicated. I found it an engaging read. I do, however, have several important observations that I believe the authors should address to further strengthen the work.

We thank the reviewer for positive and encouraging comments on the manuscript.

Most of the concerns regarding this article stem from the limited information provided to evaluate the extent of bias introduced, particularly given the impossibility of randomization in the sample selection. This last consideration is a common issue in some attempts to establish population metrics, and it is well justified in this case in the text.

We thank the reviewer for this thoughtful observation. We acknowledge that convenience sampling introduces potential selection bias. To mitigate this, participants were recruited from diverse settings across the West Bank (community centers and social media platforms) to enhance demographic and geographic representation within the constraints of political instability and restricted institutional access. As the reviewer notes, such limitations are common in conflict-affected settings where probability-based sampling is not feasible, as justified in the Methods section. These limitations are clearly outlined in the manuscript, and we emphasized that the findings can’t be generalized to the broader Palestinian population.

Although the final sample size is more than double the initially estimated target, the risk of substantial bias remains considerable. The authors acknowledge several potential sources of bias in their limitations section; however, they do not provide sufficient data to assess the magnitude of these issues, among others.

We thank the reviewer for this comment. we agree with the reviewer that a large sample size does not eliminate the risk of bias. Formal estimation of bias was not possible because the study relied on non-probability sampling and no population registry or sampling frame was available for comparison with non-participants. This limitation is now clarified in the Limitations section, and findings are interpreted cautiously.

• Participation bias. The sampling strategy appears to favor individuals with auditory problems, which could significantly distort the representativeness of the findings. This potential bias is not discussed when comparing the results with other studies that may have relied on more representative sampling strategies.

Response: We agree that participation bias is a potential limitation. Participation bias is discussed in the limitations section of the revised manuscript.

• Access-related participation bias. A greater likelihood of participation among individuals with easier access to digital platforms introduces another layer of bias, potentially confounding, in an epidemiological fashion, the reported associations and limiting the external validity of the findings.

Response: We agree that individuals' accessibility to digital platforms might be variable, and this bias is also added to the limitations section in the revised manuscript.

• Lack of quality control. No procedures are described to verify the identity or characteristics of respondents. While this is a recognized challenge in open surveys, it should be explicitly addressed, particularly given that the study population is narrowly defined in terms of age.

Response: We agree that the lack of identity confirmation may impact the participants' recruitment quality control, even though it may facilitate participants' approval to participate. We have included this in the limitations section. Identity confirmation was not applicable in our study setting; the precision of age-specific findings requires future studies with stricter verification protocols.

• Reflection about the instrument. Although I was not previously familiar with this instrument, it appears to have been widely applied across different settings. However, several methodological aspects require clarification. Was the instrument reverse-translated to ensure linguistic accuracy? Was a pilot study conducted prior to full implementation? In addition, given the length of the questionnaire, how did the authors account for the potential bias arising from respondent fatigue, whereby answers toward the end may be less accurate or consistent?

Response: To minimize respondent fatigue, the questionnaire was structured to allow participants to pause and resume, reducing the likelihood of inconsistent responses toward the end.

The questionnaire was an Arabic translation version of ESIT-SQ, which was used as a study tool in a recent Palestinian study, the questionnaire underwent a forward-backward translation process to ensure language accuracy in prior research.

A pilot study was executed before the comprehensive data collection to evaluate the clarity, relevance, and feasibility of the questionnaire. The pilot research data were excluded from the final analysis.

• Survey outreach. An estimation of the total population reached through different recruitment strategies would be valuable for evaluating coverage and representativeness.

• Attrition and completion rates. The manuscript does not report the number of participants who completed the survey relative to the number of attempts. It is implausible that all individuals who began the survey completed it. Similarly, it is unclear whether all data fields were fully completed, or whether the survey platform required mandatory responses to proceed. Clarifying this aspect would be essential, as attrition rates and missing data patterns are key indicators of data quality and potential bias.

Response: To prevent attrition and incomplete responses, all fields of the manuscript were mandatory, and only complete responses could be submitted. Thus, the response rate calculation was not applicable.

In summary, while the sample size achieved is commendable, the absence of detailed information on recruitment, response validation, and attrition undermines confidence in the robustness and representativeness of the findings.

The presentation of results is excessively detailed, which diminishes the reader’s ability to discern the specific contributions of the article. Given that the topic is described as multifactorial and that the supporting evidence for associations with diverse risk factors remains generally weak, the simultaneous testing of a large number of variables is problematic. Moreover, some analyses based on the absence of associations within the dataset are presented but not adequately developed in the discussion. I recommend that the authors streamline their results by emphasizing those factors that appear most compelling or most closely aligned with the evidence introduced in the background, thereby strengthening the coherence and contribution of the manuscript.

We appreciate the reviewer's suggestion. Given the exploratory nature of the study and the topic's varied background, we attempted to maintain a full presentation of data to ensure transparency and prevent selective reporting. Both significant and non-significant findings were considered instructive and relevant to current research. The Discussion section concentrates on the factors that are most closely related to earlier evidence, allowing for easier interpretation while keeping the overall results.

Related to the previous point and regarding to the limited robustness of the sample, the use of modeling strategies may be problematic. The article would benefit from adopting a more explicitly exploratory tone, consistently avoiding the implication that the findings are representative of the entire population. For example, the results should be systematically framed as applying in “this sample of young Palestinians”, or something similar, rather than presented as generalizable estimates. In this light, the modeling exercises can still be considered valuable, but their contribution should be interpreted as illustrative and exploratory, rather than as evidence of definitive or population-level veracity.

We completely agree that, given the sample's constraints, the study should take a more clearly exploratory approach. The phrasing in the revised manuscript has been carefully altered to minimize implications of population-level generalizability. Results are now regularly presented as applicable to "this sample of young Palestinians" rather than the general population.

There appears to be a systematic error in how prevalence percentages are calculated and communicated. For example, the second paragraph of the Discussion states that girls have a higher prevalence than boys, yet the percentages reported (32.7% and 14.3%) are inconsistent with the tabulated data. Using the authors’ own figures—730 girls in total (Table 1), of whom 370 have tinnitus (Table 6)—the prevalence among girls should be 370/730 ≈ 50.7%, not 32.7%. This suggests the denominator used was the total sample rather than the sex-specific subgroup. Please carefully review all other variables to ensure this error is not repeated, and re-evaluate whether comparisons with external studies remain valid in light of the corrected estimates.

Response: Thank you for your notice. The mentioned numbers were based on calculations using the total sample as a denominator. However, it is more logical to use the sex specific subgroup number. Numbers have been revised, and proper corrections have been made.

Reviewer #3: The partial consideration of tinnitus as a psychosocial symptom is striking, given the context, both in the introduction and the discussion. The allusion to the healthcare costs of the problem is taken out of context and is not revisited in the discussion; it might be better to remove it.

We thank the reviewer for this insightful comment. We agree that tinnitus should be conceptualized within a broader psychosocial framework. Accordingly, we have revised both the introduction and the discussion by adding a paragraph to each section that more explicitly acknowledges the psychosocial dimensions of tinnitus. (Page 3/ Introduction, Page 39/Discussion).

In addition, as healthcare costs were not analyzed in the present study and were not revisited in the discussion, we have removed the paragraph referring to the economic burden of tinnitus from the introduction to improve coherence and maintain focus on the study objectives

---

## [Decision Letter · Decision Letter 1]

29 Jan 2026

Dear Dr. ALQUB,

Thank you for submitting your manuscript to PLOS ONE. After careful consideration, we feel that it has merit but does not fully meet PLOS ONE’s publication criteria as it currently stands. Therefore, we invite you to submit a revised version of the manuscript that addresses the points raised during the review process.

We look forward to receiving your revised manuscript.

Kind regards,

Paul H Delano, Ph.D.

Academic Editor

PLOS One

Journal Requirements:

Reviewers' comments:

Reviewer's Responses to Questions

**Comments to the Author**

Reviewer #2: (No Response)

Reviewer #3: All comments have been addressed

2. Is the manuscript technically sound, and do the data support the conclusions?

Reviewer #2: Yes

Reviewer #3: Partly

3. Has the statistical analysis been performed appropriately and rigorously?

Reviewer #2: Yes

Reviewer #3: Yes

4. Have the authors made all data underlying the findings in their manuscript fully available?

Reviewer #2: Yes

Reviewer #3: Yes

5. Is the manuscript presented in an intelligible fashion and written in standard English?

Reviewer #2: Yes

Reviewer #3: Yes

Reviewer #2: I’d like to thank the authors for their thoughtful revision of the text. This revised version shows clear effort to improve the manuscript, and several points raised in the initial review have been addressed in a satisfactory manner.

I appreciate the clearer acknowledgment of the study’s exploratory nature and tone, the more explicit discussion of limitations related to the sampling strategy, and the expanded description of the questionnaire (i.e. prior validation, pilot testing)

Some other comments appear to have been addressed mainly at a declarative level. While additional limitations are now mentioned, their implications are not always incorporated into the interpretation of results or comparisons with other studies. This does not detract substantially from the manuscript.

However, I would like to draw attention to one specific issue that remains unresolved and requires further careful revision. Although the authors state that prevalence calculations were corrected, the revised manuscript continues to report sex-specific prevalence percentages using the total sample as the denominator rather than sex-specific denominators. For instance, the proportion of females with tinnitus is still reported as 32.7% (370/1131) in both the abstract and Table 6, whereas the sex-specific prevalence would be approximately 50.7% (370/730), as correctly reflected in the discussion section.

While reporting percentages using the total sample as the denominator in a “2×2 table” format can be acceptable if clearly justified and consistently communicated, the use of these percentages in the abstract is potentially misleading. Moreover, when comparing the tables in the revised manuscript with those in the original version, I do not observe any changes to the numerical values. This suggests that, beyond revisiting the calculations in the tables themselves, it is important to carefully review all related interpretations and summaries of the data to ensure consistency and accuracy throughout the manuscript.

While this appears to be a technical oversight, correcting it is important, as it affects subgroup comparisons and the interpretation of the findings. I therefore encourage the authors to revisit all prevalence calculations, ensure that appropriate denominators are used throughout, and update any related interpretations accordingly.

Overall, the manuscript has improved. and addressing this remaining issue would further strengthen its internal consistency and clarity.

Reviewer #3: Two comments regarding the insertions made in the introduction and conclusions, in response to my observations on the first version of the manuscript. (1) Highlighting the greater (but marginal) risk attributed to the age of 17 years is delicate. From a developmental perspective, adolescent development encompasses different classifications; one of the most traditional distinguishes adolescence into early, middle, and late age ranges. The article with which this study is compared and which purportedly supports its findings (ref. 20) compares age ranges (11–14 years and 14–18 years) rather than single-year ages. Moreover, the single-year ages of 15 and 17 years, which are used as points of comparison in the present study, are included within the same age range of middle adolescence from a developmental standpoint, both evolutionarily and in the cited study. This is relevant because the use of age ranges is developmentally grounded in the understanding that adolescents are undergoing continuous development, without absolute thresholds, within a phase of the life cycle that has been relatively understudied and whose upper age limit has been extended (Baird, S., Choonara, S., Azzopardi, P. S., Banati, P., Bessant, J., Biermann, O., Capon, A., Claeson, M., Collins, P. Y., De Wet-Billings, N., Dogra, S., Dong, Y., Francis, K. L., Gebrekristos, L. T., Groves, A. K., Hay, S. I., Imbago-Jácome, D., Jenkins, A. P., Kabiru, C. W., … Viner, R. M. (2025)). In addition, from a technical standpoint, the statistical test applied to the age variable in the present study evaluates the statistical hypothesis of difference versus no difference across the set of ages as a whole, rather than differences between specific single-year ages. (2) Reference 58 acknowledges contextual elements, including collective violence (I suggest review : World Health Organization. (2004). Preventing violence: A guide to implementing the recommendations of the World report on violence and health (ISBN: 9241592079). World Health Organization. Page 1) as a risk factor for PTSD, in which tinnitus would be an associated symptom of a broader clinical condition. In light of this background, I would like to invite the authors to think critically on the investigative, ethical, and epistemic perspective that challenges us all to question the value, energy, and level of detail involved in studying tinnitu as a specific symptom in relation to the adverse childhood experiences (ACEs or Adverse Childhood Events) that adolescents—understood as members of the species in a critical developmental phase that shapes their personal and societal futures—are experiencing in contexts that would make it desirable to consider variables not captured by standardized instruments developed in other settings, as frequently occurs in countries with levels of development different from those considered “developed.”

**Do you want your identity to be public for this peer review?** For information about this choice, including consent withdrawal, please see our Privacy Policy

Reviewer #2: No

Reviewer #3: No

---

## [Author Response · Author response to Decision Letter 2]

18 Feb 2026

Reviewer #2: I’d like to thank the authors for their thoughtful revision of the text. This revised version shows clear effort to improve the manuscript, and several points raised in the initial review have been addressed in a satisfactory manner.

I appreciate the clearer acknowledgment of the study’s exploratory nature and tone, the more explicit discussion of limitations related to the sampling strategy, and the expanded description of the questionnaire (i.e. prior validation, pilot testing)

Some other comments appear to have been addressed mainly at a declarative level. While additional limitations are now mentioned, their implications are not always incorporated into the interpretation of results or comparisons with other studies. This does not detract substantially from the manuscript.

However, I would like to draw attention to one specific issue that remains unresolved and requires further careful revision. Although the authors state that prevalence calculations were corrected, the revised manuscript continues to report sex-specific prevalence percentages using the total sample as the denominator rather than sex-specific denominators. For instance, the proportion of females with tinnitus is still reported as 32.7% (370/1131) in both the abstract and Table 6, whereas the sex-specific prevalence would be approximately 50.7% (370/730), as correctly reflected in the discussion section.

While reporting percentages using the total sample as the denominator in a “2×2 table” format can be acceptable if clearly justified and consistently communicated, the use of these percentages in the abstract is potentially misleading. Moreover, when comparing the tables in the revised manuscript with those in the original version, I do not observe any changes to the numerical values. This suggests that, beyond revisiting the calculations in the tables themselves, it is important to carefully review all related interpretations and summaries of the data to ensure consistency and accuracy throughout the manuscript.

Response: The percentages were revised and corrected using the sex-specific denominators rather than the total sample.

While this appears to be a technical oversight, correcting it is important, as it affects subgroup comparisons and the interpretation of the findings. I therefore encourage the authors to revisit all prevalence calculations, ensure that appropriate denominators are used throughout, and update any related interpretations accordingly.

Overall, the manuscript has improved. and addressing this remaining issue would further strengthen its internal consistency and clarity.

Reviewer #3: Two comments regarding the insertions made in the introduction and conclusions, in response to my observations on the first version of the manuscript.

(1) Highlighting the greater (but marginal) risk attributed to the age of 17 years is delicate. From a developmental perspective, adolescent development encompasses different classifications; one of the most traditional distinguishes adolescence into early, middle, and late age ranges. The article with which this study is compared and which purportedly supports its findings (ref. 20) compares age ranges (11–14 years and 14–18 years) rather than single-year ages. Moreover, the single-year ages of 15 and 17 years, which are used as points of comparison in the present study, are included within the same age range of middle adolescence from a developmental standpoint, both evolutionarily and in the cited study. This is relevant because the use of age ranges is developmentally grounded in the understanding that adolescents are undergoing continuous development, without absolute thresholds, within a phase of the life cycle that has been relatively understudied and whose upper age limit has been extended (Baird, S., Choonara, S., Azzopardi, P. S., Banati, P., Bessant, J., Biermann, O., Capon, A., Claeson, M., Collins, P. Y., De Wet-Billings, N., Dogra, S., Dong, Y., Francis, K. L., Gebrekristos, L. T., Groves, A. K., Hay, S. I., Imbago-Jácome, D., Jenkins, A. P., Kabiru, C. W., … Viner, R. M. (2025)). In addition, from a technical standpoint, the statistical test applied to the age variable in the present study evaluates the statistical hypothesis of difference versus no difference across the set of ages as a whole, rather than differences between specific single-year ages.

Response: Thank you for providing this important and constructive remark. We completely agree that interpreting age-related data in adolescent necessitates careful conceptual and methodological considerations. As a result, we have tempered the terminology in discussion sections to avoid the impression that age 17 represents a discrete developmental category. Instead, we refer to the data as a marginal variance observed throughout middle adolescence, rather than evidence of a discrete age-specific risk. We have also included a brief line in the Discussion to acknowledge modern perspectives that emphasize adolescence's flexibility and stretched upper border (e.g., Baird et al., 2025), emphasizing the absence of hard developmental thresholds.

(2) Reference 58 acknowledges contextual elements, including collective violence (I suggest review : World Health Organization. (2004). Preventing violence: A guide to implementing the recommendations of the World report on violence and health (ISBN: 9241592079). World Health Organization. Page 1) as a risk factor for PTSD, in which tinnitus would be an associated symptom of a broader clinical condition. In light of this background, I would like to invite the authors to think critically on the investigative, ethical, and epistemic perspective that challenges us all to question the value, energy, and level of detail involved in studying tinnitu as a specific symptom in relation to the adverse childhood experiences (ACEs or Adverse Childhood Events) that adolescents—understood as members of the species in a critical developmental phase that shapes their personal and societal futures—are experiencing in contexts that would make it desirable to consider variables not captured by standardized instruments developed in other settings, as frequently occurs in countries with levels of development different from those considered “developed.”

Response: Thank you for this comment. We agree with the importance of situating tinnitus within the broader clinical and contextual framework of adolescent health, as tinnitus may represent an associated symptom of conditions such as PTSD, particularly in contexts marked by collective violence and adverse childhood experiences (ACEs). In response, we have included this point in the discussion and the suggested reference has been included.

The sentence included in the discussion: Moreover, while tinnitus emerged as a reported symptom, it should be understood within the broader context of trauma-related conditions such as PTSD, particularly in conflict-affected areas such as Palestine, where the population, including all age groups, is frequently experiencing collective violence and adverse childhood experiences (ACEs)

---

## [Editor Report · Decision Letter 2]

19 Feb 2026

Prevalence and Associated Risk Factors of Tinnitus Among Palestinian Adolescents Aged 15–18: A Cross-Sectional Study

PONE-D-25-46042R2

Dear Dr. ALQUB,

We’re pleased to inform you that your manuscript has been judged scientifically suitable for publication and will be formally accepted for publication once it meets all outstanding technical requirements.

Kind regards,

Paul H Delano, Ph.D.

Academic Editor

PLOS One
---

## [Editor Report · Acceptance letter]

PONE-D-25-46042R2

PLOS One

Dear Dr. ALQUB,

I'm pleased to inform you that your manuscript has been deemed suitable for publication in PLOS One. Congratulations! Your manuscript is now being handed over to our production team.

Kind regards,

on behalf of

Dr. Paul H Delano

Academic Editor

PLOS One